# $\delta^{13}C$ decreases in the upper western South Atlantic during Heinrich Stadials 3 and 2

Marília C. Campos[1], Cristiano M. Chiessi[1], Ines Voigt[2], Alberto R. Piola[3,4], Henning Kuhnert[2], Stefan Mulitza[2]

[1]School of Arts, Sciences and Humanities, University of São Paulo, São Paulo, 03828-000, Brazil
[2]MARUM – Center for Marine Environmental Sciences, University of Bremen, Bremen, 28359, Germany
[3]Servicio de Hidrografia Naval (SHN), Buenos Aires, C1270ABV, Argentina
[4]Dept. Ciencias de la Atmósfera y los Océanos, FCEN, Universidad de Buenos Aires, C1428 EHA, and Instituto Franco–Argentino sobre Estudios de Clima y sus Impactos, CNRS/CONICET, C1428EGA, Argentina

*Correspondence to*: Marília C. Campos (marilia.carvalho.campos@usp.br)

**Abstract.** Abrupt millennial–scale climate change events of the last deglaciation (i.e., Heinrich Stadial 1 and the Younger Dryas) were accompanied by marked increases in atmospheric $CO_2$ ($CO_{2atm}$) and decreases of its stable carbon isotopic ratios ($\delta^{13}C$), i.e., $\delta^{13}CO_{2atm}$, presumably due to outgassing from the ocean. However, information on the preceding Heinrich Stadials during the last glacial period is scarce. Here we present $\delta^{13}C$ records from two species of planktonic foraminifera from the western South Atlantic that reveal major decreases (up to 1‰) during Heinrich Stadials 3 and 2. These $\delta^{13}C$ decreases are most likely related to millennial–scale periods of weakening of the Atlantic meridional overturning circulation and the consequent increase (decrease) in $CO_{2atm}$ ($\delta^{13}CO_{2atm}$). We hypothesise two mechanisms that could account for the decreases observed in our records, namely strengthening of Southern Ocean deep water ventilation and weakening of the biological pump. Additionally, we suggest that air–sea gas exchange could have contributed to the observed $\delta^{13}C$ decreases. Together with other lines of evidence, our data are consistent with the hypothesis that the $CO_2$ added to the atmosphere during abrupt millennial–scale climate change events of the last glacial period also originated in the ocean and reached the atmosphere by outgassing. The temporal evolution of $\delta^{13}C$ during Heinrich Stadials 3 and 2 in our records is characterized by two relative minima separated by a relative maximum. This "w–structure" is also found in North Atlantic and South American records, further suggesting that such structure is a pervasive feature of Heinrich Stadial 2 and, possibly, also Heinrich Stadial 3.

Keywords: Planktonic Foraminifera. Stable Carbon Isotopes. Heinrich Stadials. Atlantic Meridional Overturning Circulation.

## 1 Introduction

Heinrich Stadials (HS) are abrupt millennial–scale climate change events marked by an anti-phase interhemispheric temperature pattern which is usually termed the bipolar seesaw (Broecker, 1998). One widely accepted mechanism for the bipolar seesaw is related to changes in the strength of the Atlantic meridional overturning circulation (AMOC), likely caused by fresh water input into high latitudes of the North Atlantic (Mix et al., 1986; Crowley, 1992; Stocker, 1998). During HS, a

weak AMOC occurred simultaneously with cooling in the high latitudes of the surface North Atlantic (Sachs and Lehman, 1999; Bard et al., 2000), warming of the surface South Atlantic (Barker et al., 2009; Chiessi et al., 2015), a southward migration of the Intertropical Convergence Zone (ITCZ) (Arz et al., 1998; Deplazes et al., 2013), strengthening of the South American monsoon system (SAMS) (Cruz et al., 2006; Kanner et al., 2012), and an increase in atmospheric $CO_2$ ($CO_{2atm}$) (Ahn and

Brook, 2008; Ahn and Brook, 2014). This increase in $CO_{2atm}$ was accompanied by a decrease of its stable carbon isotopic composition ($\delta^{13}CO_{2atm}$), at least for some HS (Eggleston et al., 2016). It has been suggested that the origin of the $CO_{2atm}$ rise and the associated $\delta^{13}CO_{2atm}$ decrease was ocean–sourced (Schmittner and Galbraith, 2008; Anderson et al., 2009; Denton et al., 2010; Mariotti et al., 2016; Eggleston et al., 2016; Hertzberg et al., 2016). The occurrence of stable carbon isotope ($\delta^{13}C$) minima during HS1 (last deglaciation) in planktonic foraminiferal records from the Indo–Pacific Ocean, Southern Ocean, and

South Atlantic Ocean (Oppo and Fairbanks, 1989; Ninnemann and Charles, 1997; Mulitza et al., 1999; Spero and Lea, 2002) suggests that the signal originated from the ocean region most directly connected to all major oceanic basins, i.e., the Southern Ocean (Ninnemann and Charles, 1997). Under a weak AMOC, wind–driven upwelling of the Circumpolar Deep Water (CDW) in the Southern Ocean would become stronger, reducing the stratification of the Southern Ocean, and enhancing outgassing of low–$\delta^{13}C$ $CO_2$ to the atmosphere (Anderson et al., 2009; Denton et al., 2010; Tschumi et al., 2011; Bauska et al. 2016).

However, model experiments (e.g., Schmittner and Galbraith, 2008; Schmittner and Lund, 2015) and records from different ocean basins (e.g., Tessin and Lund, 2013; Lund et al., 2015; Curry and Oppo, 2005; Hertzberg et al., 2016) suggest that the increase in $CO_{2atm}$ and decrease in $\delta^{13}CO_{2atm}$ during HS1 are instead related to the weakening of the global oceanic biological pump and the consequent accumulation of $^{13}C$–depleted $CO_2$ in the upper water column. Such anomalously low–$\delta^{13}CO_2$ would then reach the atmosphere via air–sea gas exchange.

The reduction of the upper water column $\delta^{13}C$ caused by one or both of the mechanisms described above may be a common feature of other HS as well but, so far, planktonic foraminiferal $\delta^{13}C$ records corroborating this assumption often only cover the abrupt millennial–scale climate change events of the last deglaciation (i.e., HS1 and the Younger Dryas) while high temporal resolution information on the HS of the last glacial period is still scarce (Oppo and Fairbanks, 1989; Ninnemann and Charles, 1997; Spero and Lea, 2002; Hertzberg et al., 2016). Here we investigate this issue for HS3 and HS2 using planktonic

foraminiferal (*Globigerinoides ruber* white (*G. ruber* w) and *Globorotalia inflata* (*G. inflata*)) $\delta^{13}C$ data from a high temporal resolution marine sediment core (GeoB6212-1), collected near 32° S off south–eastern South America (SESA). Our data suggest that HS3 and HS2 were also marked by significant $\delta^{13}C$ decreases in the upper water column.

## 2 Regional setting

The upper water column of the study area is dominated by the southward flowing Brazil Current (BC) that forms the western

branch of the South Atlantic subtropical gyre. The BC is one of the weakest western boundary currents in the world ocean (Peterson and Stramma, 1991) carrying warm, saline and nutrient–depleted subtropical waters southward (Olson et al., 1988).

The BC originates near 10-15° S from the bifurcation of the Southern South Equatorial Current as it approaches the western slope of the Brazil Basin (Stramma et al., 1990; Peterson and Stramma, 1991). Around 38° S the BC encounters the northward flowing Malvinas Current (MC), where the opposing flows turn south–east and flow offshore, the so-called Brazil/Malvinas Confluence. The Brazil/Malvinas Confluence is characterized by intense mesoscale variability. After collision and

considerable mixing the warm–salty BC fractions flow eastward as the South Atlantic Current (Olson et al., 1988; Peterson and Stramma, 1991), while the majority of the cold fresh MC waters veer southeastward to rejoin the Antarctic Circumpolar Current.

In the study area, the BC transports Tropical Water (TW), South Atlantic Central Water (SACW) and Antarctic Intermediate Water (AAIW). TW occupies the mixed layer, i.e., the upper ca. 100 m of the water column, with a mean temperature of 20

°C and mean salinity of 36 psu (Tsuchiya et al., 1994). TW originates in the tropics-subtropics transition region by subduction, creating a subsurface salinity maximum capping the central waters (Mémery et al., 2000; Tomczak and Godfrey, 2003) (Fig. 1).

SACW occupies the permanent thermocline from ca. 100 to 500 m water depth. Its temperature ranges from 6 to 20 °C and its salinity spans from 34.6 to 36 psu (Mémery et al., 2000). Two types of SACW have been identified (Stramma et al., 2003).

The low–density type of SACW which is mainly found in the South Atlantic subtropical gyre is formed by subduction of a low–density type of Subantarctic Mode Water (SAMW) along the southern edge of the gyre (Stramma and England, 1999). The denser variety of SACW originates in the South Indian Ocean and is brought into the South Atlantic by the Agulhas Current (Sprintall and Tomczak, 1993) (Fig. 1).

Just below the permanent thermocline, AAIW occupies the water column from ca. 500 to 1200 m water depth (Stramma and

England, 1999). AAIW is characterized as a cold and low salinity water mass (Piola and Georgi, 1982; Tomczak and Godfrey, 2003). Around the southern tip of South America, AAIW originates by subduction of cold and fresh Antarctic Surface Water across the Antarctic Polar Front, and by contribution of a dense type of SAMW that originates from deep winter convection in the Subantarctic Zone (Molinelli, 1981; Naveira Garabato et al., 2009). AAIW is advected eastward through the Drake Passage by the Antarctic Circumpolar Current and turns northwards with the MC into the South Atlantic (Piola and Gordon, 1989).

Since AAIW circulation follows the anticyclonic flow of the subtropical gyre the majority of the northward flow at mid-latitude occurs in the eastern basin (McCartney, 1977; Stramma and England, 1999; Tomczak and Godfrey, 2003). However, intense mixing in the Brazil/Malvinas Confluence also leads to direct northward flow in the western South Atlantic that can, to some extent, influence the dissolved inorganic carbon $\delta^{13}C$ ($\delta^{13}C_{DIC}$) in the formation region of SACW (e.g., Piola and Georgi, 1982) (Fig. 1 and 2).

In the modern South Atlantic, the distribution of $\delta^{13}C_{DIC}$ allows the identification of its major water masses. TW and SACW show high $\delta^{13}C_{DIC}$ values of ca. 2‰. AAIW presents $\delta^{13}C_{DIC}$ values of ca. 0.7‰. NADW derives from the North Atlantic and shows $\delta^{13}C_{DIC}$ values of ca. 1‰. In the southwest South Atlantic NADW is sandwiched between Upper and Lower CDW which present $\delta^{13}C_{DIC}$ values of ca. 0.4‰ (Kroopnick, 1985). Since planktonic foraminiferal $\delta^{13}C$ reflects the $\delta^{13}C_{DIC}$ of the

ambient seawater, we use it as a proxy for the past oceanic carbon system (Mulitza et al., 1999; Spero, 1992). However, other factors such as calcification temperature, carbonate ion concentration, symbiont activity and air–sea gas exchange may also influence planktonic foraminiferal $\delta^{13}C$ with modification by vital effects (Lynch-Stieglitz et al., 1995; Spero and Lea, 1996, Spero et al., 1997; Bemis et al., 2000).

Changes in upper ocean properties and circulation patterns are also closely associated with changes in the atmospheric circulation. Positive sea surface temperature (SST) anomalies in the western South Atlantic, likely associated to changes in the strength of the AMOC (Knight et al., 2005), have been correlated with positive anomalies in the strength of the SAMS and, consequently, with the increase of precipitation over SESA (Chaves and Nobre, 2004). The SAMS and its main components – the ITCZ, the South Atlantic Convergence Zone (SACZ), and the South American Low Level Jet (SALLJ) – are the main

atmospheric drivers of the hydroclimate of tropical and subtropical SESA to the east of the Andes (Garreaud et al., 2009). The ITCZ is a global convective belt in the equatorial region, and the SACZ is an elongate NW-SE convective belt that originates in the Amazon Basin and extends southeastward above the northern portion of SESA and the adjacent subtropical South Atlantic. The SALLJ is a NW-SE humidity flux from the west Amazon Basin to the subtropical region of SESA (Zhou and Lau, 1998; Carvalho et al., 2004). This southward water vapour flux is a crucial source of precipitation to the Plata River

drainage basin (Berbery and Barros, 2002), which is a source of continental borne sediments to our core site.

## 3 Materials and methods

### 3.1 Marine sediment core

We investigated sediment core GeoB6212-1 (32.41° S, 50.06° W, 1010 m water depth, 790 cm core length) (Schulz et al., 2001) collected from the continental slope off SESA where the upper water column is under the influence of the BC, and thus

the TW and SACW (Fig. 1). This gravity core was raised at the Rio Grande Cone, a major sedimentary feature in the western Argentine Basin. As our study focuses in HS3 and HS2, we analysed the section from the bottom of the core (768 cm core depth; ca. 32 cal ka BP) up to 290 cm core depth (ca. 20 cal ka BP). Visual core inspection provided evidence for the presence of sand lenses at 330 and 368 cm core depth (Schulz et al., 2001; Wefer et al., 2001). Therefore we did not sample these depths. The section of interest of GeoB6212-1 was sampled every 2.5 cm with syringes of 10 cm$^3$. All samples were wet sieved, oven–

dried at 50 °C and the fraction larger than 150 μm was stored in glass vials for subsequent analyses.

### 3.2 Age model

The age model of core GeoB6212-1 is based on 14 AMS radiocarbon ages from planktonic foraminifera (mixed layer and thermocline species) (Table 1, Fig. 3). For each sample, we hand–picked under a binocular microscope around 10 mg of planktonic foraminifera shells from the sediment fraction larger than 150 μm. Samples were analysed at the Poznan

Radiocarbon Laboratory, Poland, and at the Beta Analytic Radiocarbon Dating Laboratory, USA (Table 1). All radiocarbon

ages were calibrated with the calibration curve IntCal13 (Reimer et al., 2013) with the software Bacon 2.2 (Blaauw and Christen, 2011). A marine reservoir correction of 400 years was applied with associated error of 100 years (Bard, 1988). All ages are reported as calibrated years before present (cal a BP; present is 1950 AD). To construct the age model we used Bayesian statistics in the software Bacon 2.2 (Blaauw and Christen, 2011). With the exception of mem.mean (set to 0.4) and acc.shape (set to 0.5), default parameters were used. Radiocarbon ages were assumed to be t-distributed with 9 degrees of freedom (t.a=9, t.b=10). Mean ages and 95% error margins were estimated from 10,000 downcore age-depth realizations at 0.5 cm resolution (Fig. 3).

### 3.3 Stable carbon isotope analyses

Around 10 tests of *G. ruber* w *sensu stricto* (Wang, 2000) within the size range 250-350 μm and 8 tests of *G. inflata* non–encrusted with 3 chamber in the final whorl (Groeneveld and Chiessi, 2011) within the size range 315-400 μm were hand–picked under a binocular microscope every 2.5 cm from 290 to 768 cm core depth. While the first species records the conditions at the top of the mixed layer (down to ca. 30 m) (Chiessi et al., 2007; Wang, 2000), the second species records the conditions at the permanent thermocline (ca. 350-400 m) (Groeneveld and Chiessi, 2011), allowing the reconstruction of the $\delta^{13}C$ signal of the TW and the SACW, respectively. The $\delta^{13}C$ analyses were performed on a Finnigan MAT 252 mass spectrometer equipped with an automatic carbonate preparation device at the MARUM – Centre for Marine Environmental Sciences, University of Bremen, Germany. Isotopic results are reported in the usual delta–notation relative to the Vienna Peedee belemnite. Data were calibrated against the house standard (Solnhofen limestone), itself calibrated against the NBS19 standard. The standard deviation of the laboratory standard was lower than 0.05‰ for the measuring period.

### 4 Results

### 4.1 Age model and sedimentation rates

Our age model covers the period between 32 and 6 cal ka BP (Table 1, Fig. 3). Sedimentation rates change markedly during this time interval with values ranging from 5 to 91 cm ka$^{-1}$. Three main peaks in sedimentation rate were identified at ca. 26, 23 and 15 and one minor peak at 11 cal ka BP. The two oldest sedimentation peaks occur within our period of interest (i.e., from ca. 32 until 20 cal ka BP) (Fig. 3). The mean temporal resolution of our $\delta^{13}C$ records is ca. 90 yr with values ranging from 28 to 195 yrs.

### 4.2 Stable carbon isotope values of *G. ruber* and *G. inflata*

The *G. ruber* w $\delta^{13}C$ record shows two long–term decreases, from ca. 32 to 28.2 cal ka BP with an amplitude of ca. 1‰, and from ca. 26.5 to 24.9 cal ka BP also with an amplitude of ca. 1‰ (Fig. 4a). These two negative long–term trends are separated from each other by an abrupt increase of ca. 1.3‰ ending at ca. 27.2 cal ka BP. Both long–term decreases were interrupted by

brief positive excursions, one from 29.3 to 29.1 cal ka BP with an amplitude of ca. 0.7‰ and another from ca. 26.2 to 25.8 cal ka BP with an amplitude of ca. 1‰. After the second long–term decrease, the $\delta^{13}C$ values of *G. ruber* w varied around 0.7‰. Both long–term negative excursions determine a pattern we refer to as "w–structure".

The *G. inflata* $\delta^{13}C$ record shows four negative excursions departing from a baseline of ca. 0.8‰ (Fig. 4b). The first occurs from ca. 31.5 to 29.3 cal ka BP with an amplitude of ca. 0.5‰, the second from ca. 28.8 to 28 cal ka BP with the same amplitude, the third from ca. 26.5 to 26.4 cal ka BP with an amplitude of ca. 0.8‰, and the forth from ca. 25.8 to 24.4 cal ka BP with an amplitude of ca. 0.9‰. Also in the $\delta^{13}C$ record from *G. inflata* two "w–structures" are present and are defined by the previously described negative excursions.

The "w–structures" as well as the $\delta^{13}C$ minima are synchronous for both species (Fig. 4).

## 5 Discussion

The synchronous "w–structures" present in the $\delta^{13}C$ records of both planktonic foraminiferal species analysed here occur in coeval with the millennial–scale event HS2 and, although the slight offset that is attributed to age model uncertainties, also with HS3 (Sarnthein et al., 2001; Goni and Harrison, 2010) (Fig. 4). Concomitantly, a weak AMOC was described based on $^{231}Pa/^{230}Th$ records from the Bermuda Rise (ODP Site 1063, 33.7° N, 57.6° W) (Lippold et al., 2009) (Fig. 5d). Both events are also marked by pulses of ice-rafted debris (IRD) (MD99-2331, 42.2° N, 9.7° W) (Eynaud et al., 2009) and by decreases in SST (SU8118 and MD952042, 37.5° N, 10.1° W) (Bard, 2000) in the north–eastern North Atlantic (Iberian Margin). The Greenland GISP2 ice core (72.6° N, 38.5° W) shows synchronous increases in $Ca^{+2}$, indicating changes in atmospheric circulation over Greenland (Mayewski et al., 1997) (Fig. 5a, b, c). It is noteworthy that the four records (i.e., Fig. 5a, b, c, d) mentioned above also show a "w–structure" during HS2, similar to the one shown in our $\delta^{13}C$ records. The IRD (Eynaud et al., 2009) and $Ca^{+2}$ (Mayewski et al., 1997) records also show a "w–structure" similar to ours during HS3.

Based on modern conditions, we expect our core site not to be significantly influenced by changes in the local nutrient content of the upper water column since the region is dominated by the oligotrophic BC, characteristic of western boundary currents, and is far from upwelling cells (Brandini et al., 2000). Thus, it is unlikely that changes in our $\delta^{13}C$ records are associated with local productive events driven by nutrient–cycle processes (Mulitza et al., 1999).

During HS, we expect warmer temperatures to have occurred in the upper water column of the western South Atlantic (Barker et al. 2009; Chiessi et al. 2015). This would trigger an increase in $\delta^{13}C$ values of the symbiont–bearing species investigated here if calcification temperature would dominate the $\delta^{13}C$ signal (Bemis et al., 2000), which is not the case (Fig. 4a). Additionally, given the lack of regional upper ocean reconstructions for carbonate ion concentration, we assume that increased $CO_{2atm}$ that is frequently associated with HS (Ahn and Brook, 2008; Ahn and Brook, 2014) would have been accompanied by a decrease in sea surface carbonate ion concentration (Broecker and Peng, 1993). This would promote an increase in the $\delta^{13}C_{DIC}$ but our records show a negative $\delta^{13}C$ anomaly (Fig. 4). Furthermore, we analysed a symbiont–bearing and a facultative–

symbiont species (i.e., *G. ruber* w and *G. inflata*, respectively) and both records show a similar pattern (Fig. 4) indicating that changes in symbiont activity can also be disregarded as a factor influencing our results (Spero et al., 1997; Bemis et al., 2000). We propose two primary mechanisms to explain our $\delta^{13}$C decreases: (i) changes in the strength of Southern Ocean deep water ventilation (detailed in section 5.1), and (ii) the weakening of the global oceanic biological pump (detailed in section 5.2).

Additionally, air–sea gas exchange may have acted as a secondary factor contributing to our $\delta^{13}$C decreases (detailed in section 5.3).

## 5.1 Millennial–scale changes: AMOC–induced strengthening of Southern Ocean deep water ventilation

A negative excursion during HS1 was described in planktonic foraminiferal $\delta^{13}$C records from the Indo–Pacific Ocean (Spero and Lea, 2002), Southern Ocean (Ninnemann and Charles, 1997), and South Atlantic Ocean (Oppo and Fairbanks, 1989).

Ninnemann and Charles (1997) suggested that the source for this signal was the Southern Ocean.

In the Southern Ocean CDW forms from mixing of NADW, Indian Deep Water (IDW) and Pacific Deep Water (PDW) and upwells to the south of the Antarctic Polar Front driven by the prevailing westerly winds (Marshall and Speer, 2012). Therefore, the $\delta^{13}$C signature of CDW (ca. 0.4‰) (Kroopnick, 1985) lies between that of NADW (ca. 1‰) (Kroopnick, 1985) and IDW/PDW (ca. 0.2 to -0.2‰) (Kroopnick, 1985) (Oppo and Fairbanks, 1987; Charles and Fairbanks, 1992). During periods

of weak AMOC the inflow of NADW to the Southern Ocean is reduced (Charles and Fairbanks, 1992), and the $\delta^{13}$C of CDW should decrease since the latter would have a relatively larger contribution from low–$\delta^{13}$C IDW and PDW (Spero and Lea, 2002).

Additionally, during periods of reduced AMOC the sub–tropical heat transport towards the north would decrease, leading to rising temperatures in the circum–Antarctic region (EPICA Community Members, 2006). Consequently, models of low

resolution suggest that the Southern Hemisphere westerlies would become stronger and shift southward, strengthening CDW upwelling (Toggweiler et al., 2006; Tschumi et al., 2011; Lee et al., 2011; Voigt et al., 2015). Increased upwelling would supply the ocean surface south of the Antarctic Polar Front with Si(OH)$_4$–rich, low–$\delta^{13}$C waters (Anderson et al., 2009; Hendry et al., 2012). Model experiments (Tschumi et al., 2011; Menviel et al., 2015; Bauska et al., 2016) corroborate this hypothesis by showing that stronger Southern Ocean upwelling would promote a weakening of the biological pump in the Southern Ocean.

Since upwelled CDW is hypothesized to be the dominant source of the upper and intermediate waters that leave the Southern Ocean (i.e., SAMW and AAIW) (Fig. 2), increased upwelling would transfer the low $\delta^{13}$C signal as well as the positive Si(OH)$_4$ anomaly to those waters (Oppo and Fairbanks, 1989; Ninnemann and Charles, 1997; Spero and Lea, 2002; Anderson et al., 2009; Hendry et al., 2012). Actually, a low–density type of SAMW contributes to SACW that spreads into the South Atlantic (Stramma and England, 1999). Additionally, AAIW also influences SACW through vigorous eddy mixing at the

Brazil/Malvinas Confluence (Piola and Georgi, 1982). These signals would then propagate through the thermocline SACW of the South Atlantic, and be transferred to the mixed layer TW by vertical exchange process (Tomczak and Godfrey, 2003).

The reduced stratification of the Southern Ocean and intensification of the upward transport of the remineralized carbon ($^{12}$C–enriched $CO_2$) stored for a long period in deep waters (Anderson et al., 2009; Denton et al., 2010; Jaccard et al., 2016; Mariotti et al., 2016) would increase $CO_{2atm}$ (Siple Dome, 81.7° S, 148.8° W) (Ahn and Brook, 2014) (Fig. 5j). Despite the low temporal resolution, Eggleston et al.'s (2016) Antarctic $\delta^{13}CO_{2atm}$ record shows a decrease during HS2. However, the $CO_{2atm}$ peaks occur ca. 1 ka later than the initiation of the $\delta^{13}$C decrease in our records. Spero and Lea (2002) also observed a similar offset between the increase in $CO_{2atm}$ and the decrease in Pacific Ocean planktonic foraminifera $\delta^{13}$C during HS1, and attributed this apparent offset to uncertainties in the age models of their records.

Therefore, the negative excursions in our $\delta^{13}$C records could be related to the transfer of a preformed $\delta^{13}$C signal from the subantarctic zone to the western South Atlantic via central and thermocline waters.

## 5.2 Millennial–scale changes: AMOC–induced weakening of the biological pump

Recent model experiments (e.g. Schmittner, 2005; Schmittner and Galbraith, 2008) have shown that AMOC slowdown events may cause a decrease in the global efficiency of the oceanic biological pump, being an important driver for the oceanic $CO_2$ outgassing within HS1 during the last deglaciation and possibly also during other HS, including HS3 and HS2.

NADW has low preformed nutrient waters because it is formed by nutrient depleted surface waters, where the biological pump has efficiently removed nutrients from surface waters (Marinov et al., 2008). AABW has high preformed nutrient waters because it is formed by nutrient–enriched Southern Ocean surface waters (nutrients have not being efficiently removed from surface waters). However, during weak AMOC two factors may alter the nutrient distribution and the global oceanic biological pump (Schmittner and Galbraith, 2008). First, the reduction in the NADW formation decreases the input of low preformed nutrient (high $\delta^{13}C_{DIC}$) waters to the ocean interior which becomes more dominated by high preformed nutrient (low $\delta^{13}C_{DIC}$) southern component waters (e.g., AABW). Second, the reduction of the Southern Ocean stratification induced by the decrease of salt input via NADW formation promotes the strengthening of the upwelling and subsequent sinking of high preformed nutrient (low $\delta^{13}C_{DIC}$) waters to the ocean interior, thus reducing the capacity of those unutilized nutrients to sequester carbon via the biological pump. The two factors acting in conjunction are thought to be responsible for the simulated weakening of the global efficiency of the biological pump, as well as for the increase in $CO_{2atm}$ and decrease in $\delta^{13}CO_{2atm}$ (Schmittner and Galbraith, 2008; Schmittner and Lund, 2015; Hertzberg et al., 2016).

Schmittner and Lund (2015) show that the modeled weakening of the biological pump, induced by an AMOC slowdown, reduces the ability of the surface ocean to biologically sequester isotopically light organic carbon ($^{12}$C), producing a decrease in the surface ocean $\delta^{13}C_{DIC}$ and an increase of the intermediate ocean $\delta^{13}C_{DIC}$ (lower remineralization rate). For HS1, planktonic and benthic foraminiferal $\delta^{13}$C records (Tessin and Lund, 2013; Lund et al., 2015; Curry and Oppo, 2005; Hertzberg et al., 2016) from the western South Atlantic (ca. 27°S) agree with the model output by showing a decrease in $\delta^{13}$C in the upper water column (SACW) and an increase at intermediate water depths (AAIW). Thus, the weakening of the global oceanic biological pump and consequent negative anomaly of the $\delta^{13}C_{DIC}$ in the upper water column should be captured by the tests of

planktonic foraminifera $\delta^{13}C$ during calcification (Spero and Lea, 1996; Bemis et al., 2000). The negative $\delta^{13}C_{DIC}$ during HS3 and HS2 revealed by our planktonic foraminifera provide the first observational evidence supporting the modeling results. Additionally, this mechanism also provides a possible explanation for the larger negative $\delta^{13}C$ anomaly in *G. ruber* w (mixed layer dwelling) relative to the anomaly in *G. inflata* (permanent thermocline dwelling) (Fig. 4).

It is noteworthy that the mechanism described in section 5.1, although based on a different driver for the decrease in $\delta^{13}C$, also suggests that the decreases in $\delta^{13}C$ of planktonic foraminifera from the South Atlantic would be carried by SACW (inherited from its precursor, SAMW) and thus both mechanisms (described in section 5.1 and here) are in this regard not mutually exclusive. However, the mechanism described in the present section goes against the assumption that weakening of the biological pump is related to stronger upwelling in the Southern Ocean, and that the Southern Ocean would be the source of
the low $\delta^{13}C$ signal for the South Atlantic (Lund et al., 2015; Hertzberg et al., 2016).

## 5.3 Millennial–scale changes: the role of air–sea gas exchange

The $\delta^{13}C_{DIC}$ of the surface ocean can also be affected by air–sea gas exchange (Oppo and Fairbanks, 1989; Charles and Fairbanks, 1990; Lynch-Stieglitz et al., 1995). Although this process tends towards isotopic equilibrium (especially in subtropical gyres, because the longer water residence time in these regions), the $CO_2$ exchange between the ocean and the
atmosphere does not lead to equilibrium because $CO_2$ uptake and emission will still occur in different regions and the movement and replacement of surface waters is faster than required for equilibration (Lynch-Stieglitz et al., 1995). Since the $\delta^{13}CO_{2atm}$ is lighter than $\delta^{13}C_{DIC}$, at areas of ocean $CO_{2atm}$ uptake (i.e., water mass formation regions) air–sea gas exchange has the potential to deplete $\delta^{13}C_{DIC}$ (Lynch-Stieglitz et al., 1995). Additionally, the isotopic fractionation is inversely correlated with temperature.

Therefore, we cannot exclude the possibility that the likely decrease in $\delta^{13}CO_{2atm}$ during AMOC slowdown events (Eggleston et al. 2016) (e.g., HS2) could have affected the $\delta^{13}C_{DIC}$ via air–sea gas exchange, especially in regions of water mass formation. The formation region of SACW is an area of ocean $CO_2$ uptake and may contribute to the $\delta^{13}C$ anomalies observed in our *G. inflata* record (Fig. 4b). Additionally, since the isotopic fractionation during air–sea gas exchange is temperature–dependent the weakening of the AMOC and subsequent warming of the upper subtropical South Atlantic (Barker et al. 2009; Chiessi et
al. 2015) could have contributed to the observed $\delta^{13}C$ anomalies both in the *G. ruber* w and in the *G. inflata* records (Fig. 4). However, the gradient is too small (-0.1‰ $\delta^{13}C$ per °C, Broecker and Maier-Reimer, 1992) to explain the whole changes observed in our records. If temperature was the dominant driver, unrealistic changes between 5 and 13 °C would be required to explain the full amplitudes of the $\delta^{13}C$ variations.

The $\delta^{18}O$ records from *G. ruber* w and *G. inflata* from our core (Supplementary material Figure 1) should partially reflect
changes in water temperature (ca. -0.22‰ per 1 °C; e.g., Mulitza et al, 2003), but show no clear trends across HS3 and HS2. While temperature changes might be partially obscured in the foraminiferal $\delta^{18}O$ records by the influence of synchronous

changes in seawater-$\delta^{18}$O, as has been hypothesized for the Holocene (Chiessi et al., 2014) and HS1 (Chiessi et al., 2015) in the western Atlantic, we consider it unlikely that temperature changes of the above magnitude would be completely masked.

## 5.4 Changes in continental climate

Paleoclimate records from South America indicate marked hydrological changes during abrupt millennial–scale climate events (Arz et al., 1998; Peterson et al., 2000; Baker et al., 2001; Cruz et al., 2006; Stríkis et al., 2015). Reconstructed SAMS activity suggests strengthening during HS (Cruz et al., 2006; Kanner et al., 2012; Cheng et al., 2013). Changes in speleothem oxygen isotopic composition from the western Amazon Basin (NAR-C, Cueva del Diamante cave, northern Peru, 5.4° S, 77.3° W) (Cheng et al., 2013) as well as changes on gamma radiation records from the Bolivian Altiplano (Salar de Uyuni, 20.3° S, 67.5° W) (Baker et al., 2001) (Fig. 5f, g) indicate increased precipitation during HS3 and HS2. North of the equator, a reflectance record from the Cariaco Basin (off northern Venezuela, MD03-2621, 10.7° N, 65° W) (Deplazes et al., 2013) suggests decreased precipitation during the same millennial–scale events (Fig. 5e). The opposite precipitation variations at these sites reflects the interhemispheric anti-phased response of tropical precipitation during HS (Wang et al., 2007; Cheng et al., 2013). During HS3 and particularly HS2 the three above mentioned records (Fig. 5e, f, g) show a "w–structure" similar to the one observed in our $\delta^{13}$C records. Stríkis et al. (2015) reported a similar "w–structure" during HS1 related to two distinct hydrologic phases within HS1.

Periods of intensified SAMS would have strengthened the discharge from the Plata River drainage basin (Chiessi et al., 2009), increasing the delivery of terrigenous sediments to the Rio Grande Cone (Lantzsch et al., 2014), our coring site. Our records present for the first time increased sedimentation rates during a HS off SESA and corroborate the suggestion of Chiessi et al. (2009) during HS2. Furthermore, GeoB6212-1 sedimentation rates also show a "w–structure" during HS2 (Fig. 3), hinting for a sensitive response of the Plata River drainage basin to the increase in activity of the SAMS. The occurrence of a similar "w–structure" in North Atlantic records, in South American records and in our $\delta^{13}$C and sedimentation rate records gives us confidence that such "w–structure" is indeed an ubiquitous feature of HS2, and possibly also HS3 (in this case, only for $\delta^{13}$C). However, other factors like shifts in bottom currents and sea level could have also produced the observed changes in sedimentation rates. Detailed age models and more cores from the Rio Grande Cone are necessary to elucidate the main factors controlling the sedimentation rates in that region.

The increased continental runoff that led to increased delivery of terrigenous sediments to our core site could have also enhanced the nutrient availability and the local primary productivity, affecting our planktonic foraminiferal $\delta^{13}$C records. Some aspects of the regional response to HS1 are useful to evaluate this possibility. During HS1, ice volume corrected seawater–$\delta^{18}$O from the upper water column of our core site indicates an increase in salinity (Chiessi et al., 2015). Thus, despite of the increased terrigenous discharge, it seems that the upper water column of our core site was not affected by an increase in freshwater discharge from the Plata River during HS1. Since the precipitation anomaly of HS1 was stronger than that of HS3 and HS2 in the Plata River drainage basin (Wang et al., 2007), it is unlikely that weaker precipitation anomalies of HS3 and

HS2 would have impacted the upper water column of our core site more intensely than during HS1. This suggests that changes in the discharge of the Plata River drainage basin at millennial–scale are not a relevant driver of our $\delta^{13}$C decreases, and that the buoyant low salinity waters were advected elsewhere by winds, while terrigenous sediments were already too deep to be influenced by the wind.

## 6 Conclusions

Our mixed layer and permanent thermocline $\delta^{13}$C records from the western South Atlantic show in-phase millennial–scale decreases of up to 1‰ during the HS3 and HS2. We hypothesize that the low $\delta^{13}$C signal may be related to two millennial–scale mechanisms. (i) Changes in the Southern Ocean deep water ventilation. A weak AMOC during HS3 and HS2 would produce stronger Southern Ocean upwelling that in turn, would supply the surface of the Southern Ocean with more low–$\delta^{13}$C

waters as well as promote increased outgassing of this old and low–$\delta^{13}$C respired $CO_2$. The low–$\delta^{13}$C waters at the surface of the Southern Ocean would be subducted into the central and thermocline waters and transferred equatorward via the South Atlantic subtropical gyre circulation and southward along the western boundary towards our core site. (ii) Weakening of the global oceanic biological pump. A weak AMOC during HS3 and HS2 would promote an accumulation of $^{13}$C–depleted $CO_2$ in the upper water column of the South Atlantic. This accumulation would result in a negative anomaly of the $\delta^{13}C_{DIC}$ (as well

as of the $\delta^{13}CO_{2atm}$) that in turn would be captured by the tests of planktonic foraminifera at our core site. We further suggest that changes in air–sea gas exchange could have contributed to the decreases in $\delta^{13}$C via both mechanisms. Together with other lines of evidence, our data are consistent with the hypothesis that the $CO_2$ added to the atmosphere during abrupt millennial–scale climate change events of the last glacial period originated in the ocean and reached the atmosphere by outgassing. Moreover, the occurrence of a similar "w–structure" during HS2 (and possibly HS3) in North Atlantic and South American

records as well as in our planktonic foraminiferal $\delta^{13}$C and sedimentation rate records gives us confidence that such "w–structure" is a pervasive feature that characterizes HS2 (and possibly HS3).

### Data availability

The data reported here will be archived in Pangaea ([www.pangaea.de](www.pangaea.de)).

### Acknowledgements

We thank Y. Zhang for help with the Bacon software. Logistic and technical assistance was provided by the captain and crew of the R/V Meteor. We thank two anonymous reviewers and A. Schmittner for constructive comments that greatly improved this manuscript. M. C. Campos acknowledges the financial support from FAPESP (grants 2013/25518-2 and 2015/11016-0), and C. M. Chiessi acknowledges the financial support from FAPESP (grant 2012/17517-3) and CAPES (grants 1976/2014 and

564/2015). H. Kuhnert, S. Mulitza and I. Voigt were funded through the DFG Research Centre/Cluster of Excellence "The Ocean in the Earth System". A. R. Piola was funded by grant CRN3070 from the Inter-American Institute for Global Change Research through the US National Science Foundation grant GEO-1128040. Sample material was provided by the GeoB Core Repository at the MARUM – Center for Marine Environmental Sciences, University of Bremen, Germany.

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

**Table 1: Accelerator mass spectrometer radiocarbon dates and calibrated ages used to construct the age model of core GeoB6212-1.**

| Core depth (cm) | Lab ID | Species | Radiocarbon age ± 1σ error (a BP) | Calibrated ages (cal a BP) |
|---|---|---|---|---|
| 8 | Poz-47236* | *G. ruber* | 5250 ± 40 | 5929 |
| 33 | 382580** | *G. ruber* | 9440 ± 40 | 10047 |
| 58 | Poz-47237* | *G. ruber* | 10250 ± 50 | 11395 |
| 113 | 382581** | Planktonic forams | 12870 ± 50 | 14342 |
| 158 | Poz-47238* | Planktonic forams | 13050 ± 70 | 15247 |
| 253 | 382582** | Planktonic forams | 15750 ± 50 | 18620 |
| 323 | 382583** | Planktonic forams | 17560 ± 60 | 20748 |
| 363 | Poz-47239* | Planktonic forams | 18610 ± 140 | 21834 |
| 478 | 382584** | Planktonic forams | 19810 ± 70 | 23537 |
| 578 | Poz-47240* | Planktonic forams | 21750 ± 150 | 25587 |
| 623 | 382585** | Planktonic forams | 22320 ± 80 | 26184 |
| 668 | 382586** | Planktonic forams | 22480 ± 80 | 26711 |
| 717 | 424077** | Planktonic forams | 24190 ± 110 | 27850 |
| 768 | 382587** | Planktonic forams | 29520 ± 160 | 31966 |

* Poz: Poznan Radiocarbon Laboratory, Poznan, Poland.
** Beta Analytic Radiocarbon Dating Laboratory, Miami, USA.

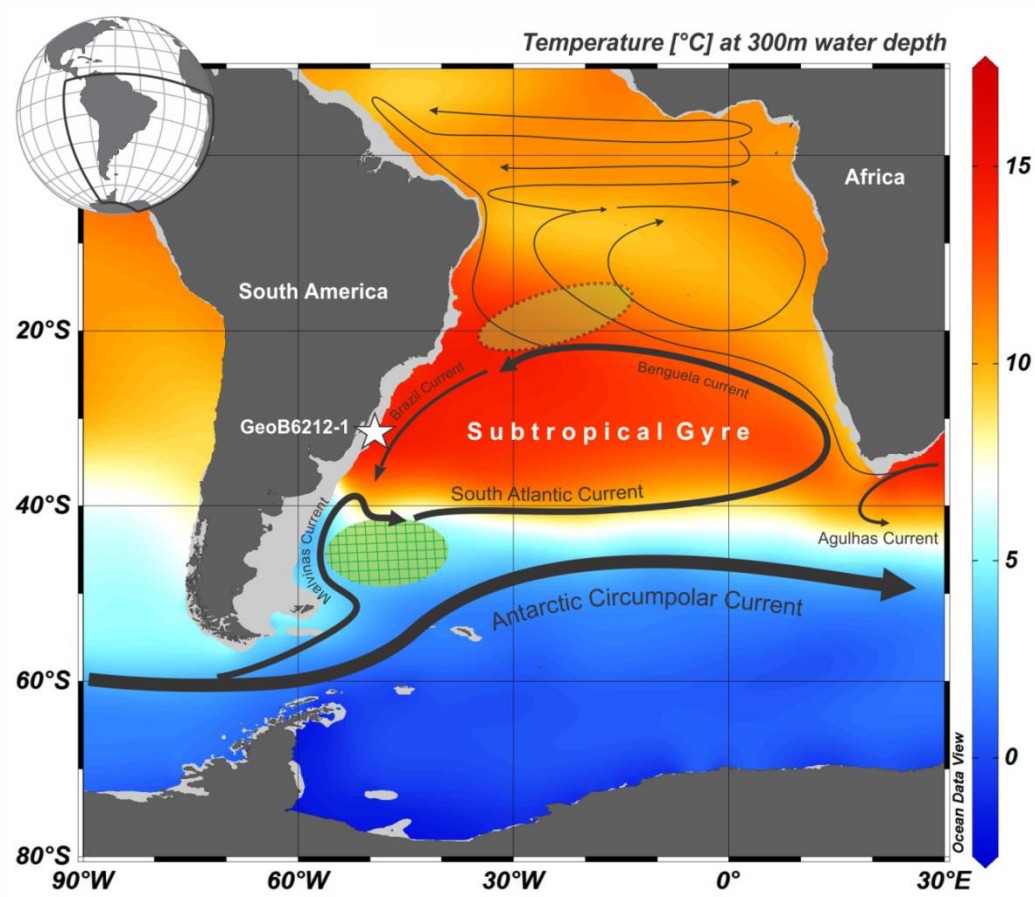

**Figure 1: Schematic representation of the large–scale circulation of South Atlantic Central Water (SACW) (Stramma and England, 1999). The main SACW source region is depicted by the gridded green ellipse, and the main source region of tropical subsurface water (TW) is indicated by the dotted yellow ellipse. Mean annual temperature at 300 m water depth is shown by the colour shading (Locarnini et al., 2013) (http://odv.awi.de).**

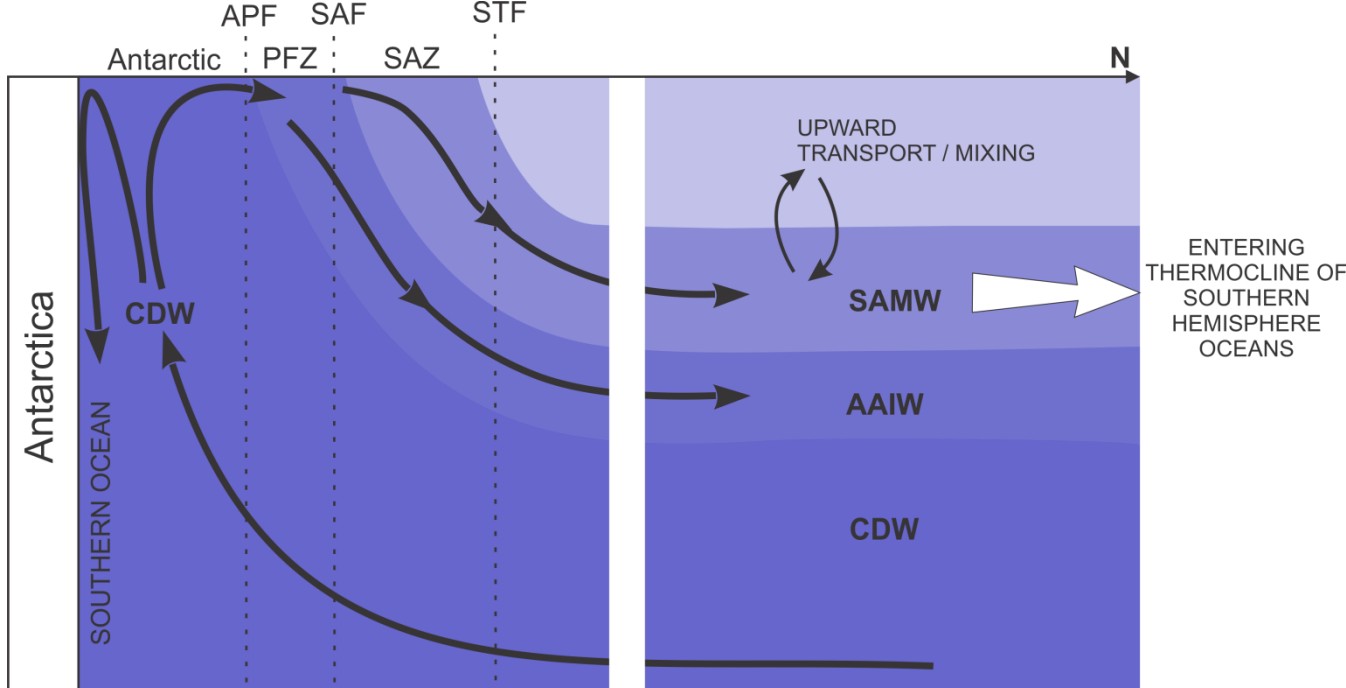

**Figure 2: Schematic representation of ventilation and subduction of water masses in the Southern Ocean (modified after Anderson et al., 2009). Wind–driven upwelling south of the latitude of maximum westerlies brings Circumpolar Deep Water (CDW) to the surface and contributes to Antarctic Surface Waters. Antarctic Surface Waters represent the dominant source of the upper and intermediate waters that leave the Southern Ocean. Antarctic Intermediate Water (AAIW) originates by subduction of cold and fresh Antarctic Surface Waters across the Antarctic Polar Front (APF) and enters the South Atlantic mainly via the subtropical gyre. Subantarctic Mode Water (SAMW) originates from deep winter convection north of the Subantarctic Front (SAF). A low–density type of SAMW enters the thermocline of the Southern Hemisphere oceans along the southern edge of the subtropical gyres where it becomes part of central waters and contributes to ventilating the thermocline, while a denser type of SAMW formed in the eastern South Pacific is regarded as a precursor of the AAIW. The Polar Front Zone (PFZ) and Subantarctic Zone (SAZ) are the regions between the APF and SAF, and between the SAF and Subtropical Front (STF), respectively.**

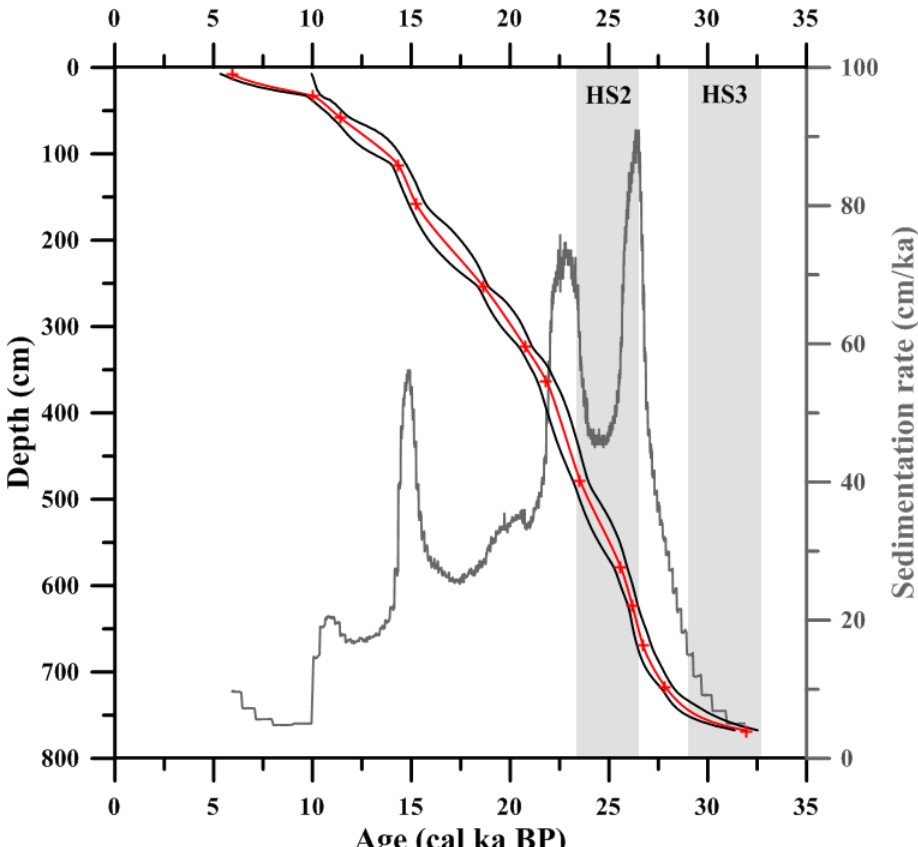

**Figure 3: Age model (left hand side y–axis; red line and black enveloping curves) and sedimentation rates (right hand side y–axis; grey line) for marine sediment core GeoB6212-1 produced with the software Bacon 2.2 (Blaauw and Christen, 2011). For the age model, the red symbols show calibrated ages, the red line depicts mean ages and the upper (lower) black line depicts maximum (minimum) ages. Grey vertical bars show abrupt millennial–scale climate change events Heinrich Stadial 3 (HS3) and Heinrich Stadial 2 (HS2) (Goni and Harrison, 2010; Sarnthein et al., 2001).**

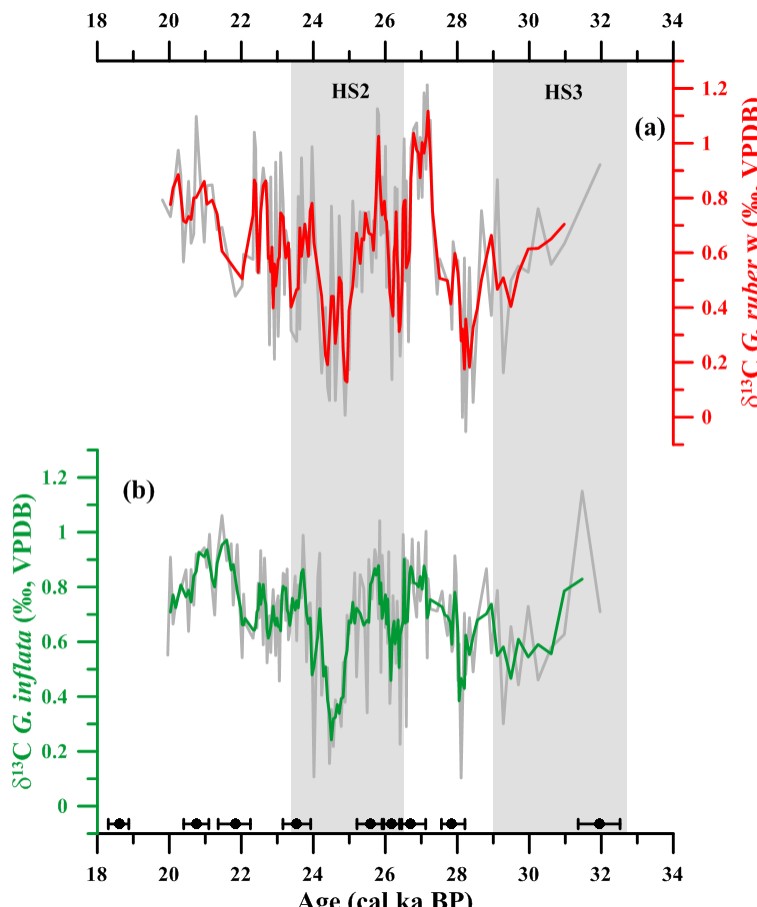

**Figure 4: Stable carbon isotopic (δ¹³C) records from sediment core GeoB6212-1. (a)** *Globigerinoides ruber* **white (*G. ruber* w) δ¹³C and (b)** *Globorotalia inflata* **(*G. inflata*) δ¹³C. Red and green lines represent three–point running averages for *G. ruber* w and *G. inflata*, respectively. Black symbols at the bottom of the panel depict calibrated ages. Grey vertical bars show abrupt millennial– scale climate change events Heinrich Stadial 3 (HS3) and Heinrich Stadial 2 (HS2) (Goni and Harrison, 2010; Sarnthein et al., 2001).**

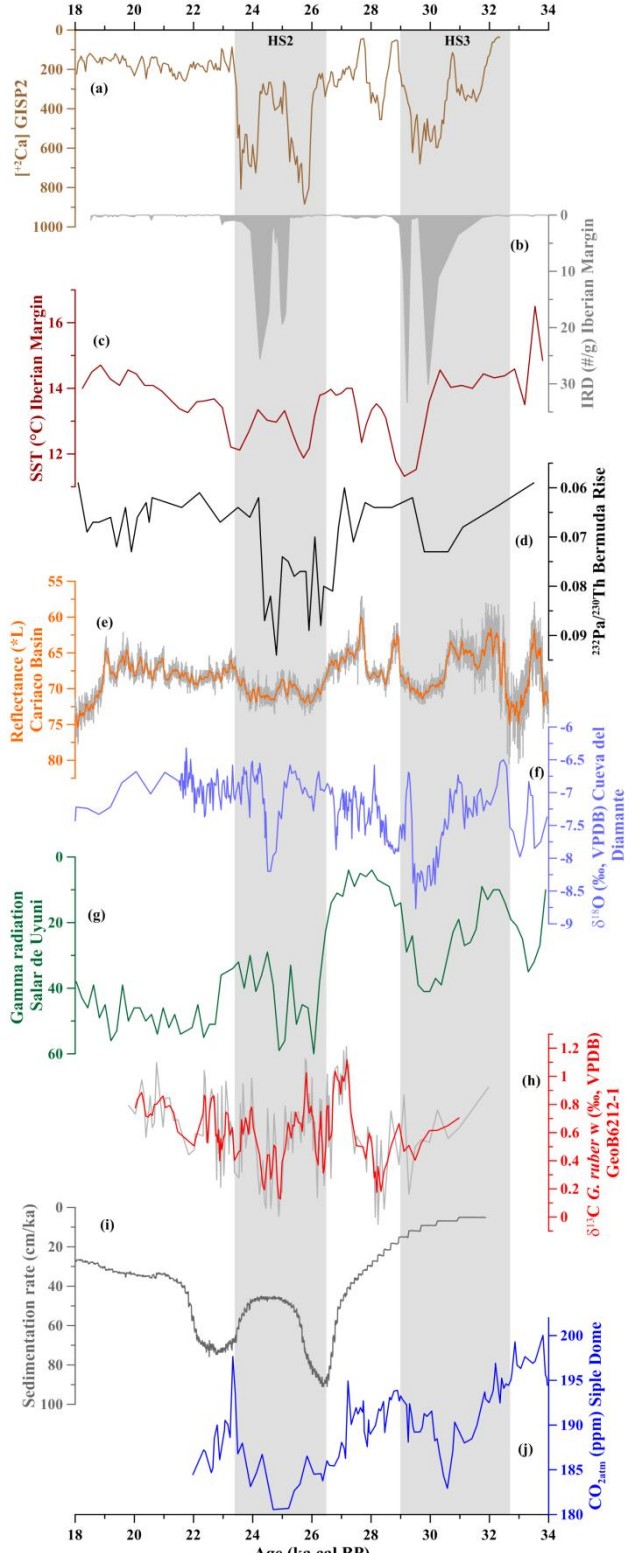

**Figure 5: Proxy records from the western South Atlantic, western and eastern North Atlantic and tropical South America spanning Heinrich Stadial 3 (HS3) and Heinrich Stadial 2 (HS2) (Goni and Harrison, 2010; Sarnthein et al., 2001). (a) Changes in atmospheric circulation over Greenland derived from Greenland Ice Sheet Project 2 (GISP2) $Ca^{+2}$ concentration (Mayewski et al., 1997) plotted versus the Greenland Ice Core Chronology 2005 (GICC05) (Andersen et al., 2006; Rasmussen et al., 2006) at 72.6° N, 38.5° W; (b) Heinrich layers indicated by the presence of ice-rafted debris (IRD) from the Iberian Margin marine sediment core MD99-2331 at 42.2° N, 9.7° W (Eynaud et al., 2009); (c) sea surface temperature (SST) (°C) changes from Iberian Margin marine sediment cores SU8118 and MD952042 at 37.5° N, 10.1° W (Bard, 2000); (d) Atlantic meridional overturning circulation (AMOC) strength derived from Bermuda Rise sedimentary $^{231}Pa/^{230}Th$ ratio – ODP Site 1063 (higher values indicate a reduced AMOC) at 33.7° N, 57.6° W (Lippold et al., 2009); (e) position of the Intertropical Convergence Zone (ITCZ) indicated by reflectance (*L) (higher values indicate decreased precipitation) from the Cariaco Basin marine sediment core MD03-2621 at 10.7° N, 65° W (Deplazes et al., 2013) (orange line represents a 399-point running average); (f) strength of western Amazon precipitation indicated by the $\delta^{18}O$ from stalagmite NAR-C collected in the Cueva del Diamante Cave, western Amazon (more negative values indicate increased precipitation) at 5.4° S, 77.3° W (Cheng et al., 2013); (g) presence of paleolakes indicated by the natural gamma radiation from Bolivian Altiplano Salar de Uyuni (higher values indicate increased precipitation) at 20.3° S, 67.5° W (Baker et al., 2001); (h) *Globigerinoides ruber* white (*G. ruber* w) $\delta^{13}C$ from marine sediment core GeoB6212-1 collected in the western South Atlantic at 32.4° S, 50.1° W (red line represents a three–point running average) (this study); (i) sedimentation rates from marine sediment core GeoB6212-1 collected in the western South Atlantic at 32.4° S, 50.1° W (this study); (j) atmospheric $CO_2$ concentration (ppm) from ice core Siple Dome (Ahn and Brook, 2014) plotted versus the Greenland Ice Core Chronology 2005 (GICC05) (Svensson et al., 2008) at 81.7° S, 148.8° W.**