# Peer review of "$\delta^{13}C$ decreases in the upper western South Atlantic during Heinrich Stadials 3 and 2"

_Climate of the Past, 2016_

## Short Comment (SC1) · 20 Jul 2016

I suggest to the authors to consider our relevant recent modeling work, which suggests a different mechanism for the CO2 increase during H-events. As shown in Schmittner and Galbraith (2008, Nature, 456, 373-376, doi:10.1038/nature07531) an AMOC shutdown causes a decrease of the efficiency of the biological pump, which leads to an increase in atmospheric CO2 consistent in both amplitude and rate-of-change with ice core observations. Schmittner and Lund (2015; Climate of the Past, 11, 135-152, doi:10.5194/cp-11-135-2015) show that this leads to a decrease of surface ocean (and atmospheric) d13C that is particularly strong (more than 0.5 permil) in the South Atlantic (their Fig. 5G).

---

## Referee Comment (RC1) · Anonymous Referee #1 · 24 Jul 2016

This is an interesting manuscript presenting two high-resolution planktic d13C records from the Brazilean coast covering HS3 and HS2. It is worth publishing in Climate of the Past if the "Discussion" section is completely rewritten and therefore the interpretation of the results re-assessed.

The authors have not shown that the planktic d13C decrease measured in their core was due to stronger Southern Ocean upwelling. This is just an hypothesis. Planktic d13C is influenced by several factors such as changes in oceanic circulation, mixing, SST, export production... (see Charles et al. 1993, Lynch-Stieglitz et al. 1995, Menviel et al. 2015...). An AMOC decrease will lead to significant surface ocean d13C changes. Changes in Southern Ocean upwelling are not the only solution. In addition, it has been shown that during calcification temperature and carbonate ion content could have an impact on calcite d13C in planktic species (see Spero et al. 1997, Bemis et al. 2000. . .). Moreover, the authors suggest that increased runoff from the Plata river drainage basin led to increased sediment rate. It is an interesting result, which nicely fits with a southward shift of the ITCZ during that time, however river runoff might potentially have a fairly low d13C signature, thus potentially also influencing surface d13C at the core location?

Please note that: 1) the ice core data (Ahn and Brook 2014) do not support any atmospheric CO2 increase during HS2 and HS3. 2) D13CO2 can't really be used due to poor resolution and most likely issues with age model. 3) The link between changes in southern hemispheric westerlies and AMOC changes is still poorly documented. 4) the opal flux in the Southern Ocean (Anderson et al. 2009) does not increase during HS2 and HS3. As such the whole discussion section, as well as conclusion and abstract need to be rewritten.

---

## Referee Comment (RC2) · Anonymous Referee #2 · 25 Aug 2016

Review of "Glacial d13C decreases in the western South Atlantic forced by millennial changes in Southern Ocean ventilation" by Campos et al.

The authors use the d13C of planktonic foraminifera to infer variability in the d13C of DIC in surface/thermocline waters in the SW Atlantic. One strength of the paper is the comprehensive review of the modern oceanographic setting. Another is that the planktonic time series are based on a sediment core with very high sedimentation rates, allowing for high resolution reconstruction of surface ocean d13C during Heinrich Stadial 2 and 3. It is also important that the authors used two different species to reconstruct d13C (one surface dwelling and another thermocline dwelling) to account for potential biases in habitat and vital effects that could overprint changes in the d13C

of DIC. Given the high resolution and replicated nature of the record, the authors clearly show that this part of the Southwest Atlantic underwent significant changes in d13C during Heinrich Stadial 2 and 3.

The primary weakness of the paper is the interpretation of the d13C and sedimentation rate results. The authors are quick to assume that their d13C records reflect the input of light carbon from the Southern Ocean and neglect other possible explanations that are well documented in the published literature. The authors also invoke speculative connections between rainfall and sedimentation rate in the core as support for their climate interpretation, despite disagreement between the sedimentation rate and planktonic d13C patterns. Finally, the authors neglect to mention much of the work at the Brazil Margin spanning the last deglaciation (including Heinrich Stadial 1) that is both relevant to their work and inconsistent with a Southern Ocean driver.

Major issues:

While the prevailing view is that Southern Ocean outgassing drove surface ocean d13C anomalies during the last deglaciation, the authors also need to reference to Andreas Schmittner's work that shows that weakening of the AMOC can alter the preformed nutrient budget of the global ocean and therefore the efficiency of the biological pump. Weakening of the biopump would preferentially leave light carbon in the surface and create negative d13C anomalies in multiple ocean basins (e.g. Schmittner, 2005; Schmittner and Galbraith, 2008). Also, detailed reconstructions from the Brazil Margin show that benthic d13C and d18O changed late in the deglaciation, suggesting that the abyssal ocean was an unlikely source for the surface ocean anomalies during HS1 (Lund et al., 2015). Additional published records from the SE Atlantic shows a similar pattern, with a late deglacial response in the abyssal records (Waelbroeck et al., 2011; Roberts et al., 2016). Furthermore, intermediate depth d13C reconstructions from the Brazil Margin (Oppo and Horowitz, 2000; Curry and Oppo, 2005; Lund et al., 2015) imply that AAIW and SAMW were not carriers of a light carbon signal during HS1, which is inconsistent with the mechanism invoked by Campos et al. While

the authors did a commendable job summarizing the literature pertaining the modern oceanographic setting, the introductory material includes surprisingly little information from previously published work along the Brazil Margin. As a result, the introductory material is incomplete and the lack of context limits the interpretation in the discussion section.

The other main issue with the data interpretation is that the continental hydroclimate response isn't well supported by the sedimentation rate data. While sedimentation rates peak early in HS2, at about the same time as a planktonic d13C minimum, there is a second peak in sedimentation rate after HS2 that corresponds to a broad maximum in d13C. Furthermore, there is no peak in sedimentation rate during either HS3 or HS1. Taken as a whole, the sedimentation rate data therefore do not support a continental hydroclimate connection, which perhaps isn't surprising given the many factors that can influence sedimentation rates at a single core site.

Finally, the manuscript would benefit from inclusion of the planktonic d18O data. If there is a clear hydroclimate signal at this site, it could appear in the d18O data as intervals of unusually low d18O. The d18O data would also be informative for assessing the influence of air-sea gas exchange effects on the d13C signal. The d13C of surface ocean DIC is influenced by temperature dependent gas exchange, with a relationship of $\sim$ -0.1 per mil per deg C of warming (Broecker and Maier-Reimer, 1992; Lynch-Stieglitz et al., 1995). While such an effect is unlikely to explain the full magnitude of the d13C signals, it should be included in the discussion. Model results suggest that weakening of the AMOC warms the upper subtropical South Atlantic by 2-3 deg C (Marcotte et al., 2011) which would account for up to 0.3 per mil of the planktonic d13C anomalies.

Minor issues:

Page 6, Line 12: Here the authors should consider the possibility that weakening of the biological pump could have produced d13C anomalies not only locally, but on a larger

spatial scales.

Page 6, Line 20: It is important to mention that published mode and intermediate water records from the Brazil Margin either show a positive d13C anomaly during HS1 or a delayed negative d13C response that is inconsistent the light carbon being transported northward via mode and intermediate water.

Page 7, Line 5: The authors seem to take it as a given that Southern Hemisphere westerlies drives greater upwelling of deep waters when it actuality there is limited data to support such a link. Please qualify these types of statements to reflect the limited constraints that exist.

Page 7, Line 9: See comments above that published intermediate depth records from the Brazil Margin don't show clear negative carbon isotope anomalies during HS1, which is relevant to whether a similar process occurred during HS2 and HS3.

Figure 4: If the authors are going to pursue the sedimentation rate-hydoclimate link, then the sedimentation rates should also be included in this figure for comparison to the d13C. The d18O records should also be included to assess the potential impact of sea surface temperature on d13C.

Figure 5: There are several time series in this plot that aren't essential for the discussion, such as the Iberian Margin SST record, the EDML d18O time series, and the Siple Dome CO2.

---

## Author Comment (AC2) · 14 Oct 2016

We thank Referee #1 for the constructive review of our manuscript. To facilitate the discussion, we have copied his/her comments below in black and inserted our responses in green.

This is an interesting manuscript presenting two high-resolution planktic d13C records from the Brazilean coast covering HS3 and HS2. It is worth publishing in Climate of the Past if the "Discussion" section is completely rewritten and therefore the interpretation of the results re-assessed.

The authors have not shown that the planktic d13C decrease measured in their core was due to stronger Southern Ocean upwelling. This is just an hypothesis.

Comment #1 - We agree that it is a hypothesis and have now treated it accordingly by rephrasing specific parts of the revised version of our manuscript.

Considering this comment and the similar suggestions from Referee #2 and Dr. Schmittner, we also incorporated two additional hypotheses (see below) in the revised manuscript, in order to have a more thorough and balanced discussion:

 i. AMOC–induced weakening of the biological pump

As shown by model experiments (e.g., Schmittner and Galbraith, 2008; Schmittner and Lund, 2015), AMOC slowdown events may cause a decrease in the global efficiency of the oceanic biological pump, being an important driver for the oceanic $CO_2$ outgassing during the last deglaciation, i.e., HS1 (but most likely also during other HS, e.g., HS3 and HS2). Two factors acting in conjunction are claimed to be responsible for the simulated weakening of the efficiency of the biological pump (Schmittner and Galbraith, 2008). First, the reduction in North Atlantic Deep Water (NADW) formation decreases the input of low preformed nutrient (high $\delta^{13}C_{DIC}$ values) waters to the interior ocean which becomes more dominated by the high preformed nutrient southern component (low $\delta^{13}C_{DIC}$ values) waters (e.g., Antarctic Bottom Water and Antarctic Intermediate Water). Second, the reduction of the Southern Ocean stratification induced by the decrease of salt input via NADW formation promotes the strengthening of the upwelling and subsequent sinking of high preformed nutrient (low $\delta^{13}C_{DIC}$ values) waters to the interior ocean, thus reducing the capacity of those unutilized nutrients to sequester carbon via the biological pump.

The weakening of the biological pump would promote the accumulation of isotopically light organic carbon in the upper water column of the South Atlantic and the negative anomaly of the $\delta^{13}C_{DIC}$ as well as of the $\delta^{13}CO_{2atm}$ (Schmittner and Lund, 2015). This $\delta^{13}C_{DIC}$ anomaly should be captured by the tests of planktonic foraminifera $\delta^{13}C$ because they incorporate $\delta^{13}C_{DIC}$ during calcification (Spero and Lea, 1996; Bemis et al., 2000). Our planktonic foraminifera $\delta^{13}C$ records agree with this hypothesis by showing negative anomalies during HS3 and HS2. Additionally, this mechanism also provides a possible explanation for the larger negative $\delta^{13}C$ anomaly in *G. ruber* w

(mixed layer–dwelling) relative to the anomaly in *G. inflata* (permanent thermocline–dwelling).

ii.    the role of air–sea gas exchange

In the original version of our manuscript (page 7, line 13) we briefly mentioned the role of air–sea gas exchange. However, we see the need to further expand this topic in order to have a more thorough and balanced discussion.

We are aware that the $\delta^{13}C_{DIC}$ of the surface ocean can be affected by air–sea gas exchange (Oppo and Fairbanks, 1989; Charles and Fairbanks, 1990; Lynch-Stieglitz et al., 1995). Therefore, we cannot exclude the possibility that the likely decrease in $\delta^{13}CO_{2atm}$ during AMOC slowdown events (Eggleston et al., 2016) (e.g., HS3 and HS2) could have affected the $\delta^{13}CO_{2DIC}$ via air–sea gas exchange, especially in regions of water mass formation. The formation regions of South Atlantic Central Water and Tropical Water are areas of major ocean $CO_2$ uptake and may contribute to the $\delta^{13}C$ anomalies observed in our records. Additionally, since the isotopic fractionation during air–sea exchange is temperature–dependent, the weakening of the AMOC and subsequent warming of the upper subtropical South Atlantic (Barker et al. 2009; Chiessi et al. 2015) could also have contributed to the observed $\delta^{13}C$ anomalies in our records (as suggested by Referee #2).

However, for the reasons stated below we still consider the possible role played by changes in Southern Ocean ventilation in producing our $\delta^{13}C$ anomalies:

i.    the mechanisms that invoke the Southern Ocean as one of the $CO_{2atm}$–sources during the last deglaciation are based on reconstructions as well as conceptual and numerical models that, to our understanding, have not yet been negated (Toggweiler et al., 2006; Anderson et al., 2009; Sigman et al., 2010; Tschumi et al., 2011; Lee et al., 2011; Burke and Robinson, 2012);

ii.    model experiments that question the role of the Southern Ocean (e.g., Schmittner and Lund, 2015) were performed under preindustrial boundary conditions, which can significantly influence the results (e.g., Menviel et al., 2008), and do not constitute a firm negation of the Southern Ocean hypothesis.

Planktic d13C is influenced by several factors such as changes in oceanic circulation, mixing, SST, export production: : : (see Charles et al. 1993, Lynch-Stieglitz et al. 1995, Menviel et al. 2015: : :).

Comment #2 - We agree that planktonic foraminiferal $\delta^{13}C$ can be affected by several factors and that some of them where not thoroughly considered in the original version of our manuscript. To account for these different factors we have inserted two new sections (i.e., (i) AMOC–induced weakening of the biological pump, and (ii) the role of air–sea gas exchange) in our manuscript (please see our Comment #1 above) and clearly refer to the complexity of the proxy.

Additionally, by analyzing two different species of planktonic foraminifera, i.e., *G. ruber* w that is a symbiont–bearing species, and *G. inflata* that is a symbiont–facultative species (e.g., Hemleben et al., 1989), we have excluded the potential bias in our records related to changing symbiont activity (e.g., Spero and Lea, 1993; Spero et al., 1997; Bemis et al., 2000). Since the $\delta^{13}$C record of both species show a similar pattern, changing symbiont activity can be disregarded as a potential bias to our results.

An AMOC decrease will lead to significant surface ocean d13C changes. Changes in Southern Ocean upwelling are not the only solution.

Comment #3 - Agreed. As mentioned in our Comment #1 (please see above) we have revised our manuscript accordingly by incorporating two additional mechanisms (i.e., (i) AMOC–induced weakening of the biological pump, and (ii) the role of air–sea gas exchange) that could account for the $\delta^{13}$C changes in our records.

In addition, it has been shown that during calcification temperature and carbonate ion content could have an impact on calcite d13C in planktic species (see Spero et al. 1997, Bemis et al. 2000: : :).

Comment #4 - Experiments with *O. universa* (a symbiont–bearing species) and *G. bulloides* (a non–symbiont species) suggest that under higher temperatures the first species presents higher $\delta^{13}$C values whereas the second species presents lower $\delta^{13}$C values (Bemis et al., 2000). These opposed $\delta^{13}$C responses were related to the presence (i.e., *O. universa*) or absence (i.e., *G. bulloides*) of symbionts. We analyzed a symbiont–bearing and a facultative–symbiont species (i.e., *G. ruber* w and *G. inflata*, respectively), but both records showed a similar behavior. During HS, we would expect warmer temperatures in the upper water column of the western South Atlantic (Barker et al. 2009; Chiessi et al. 2015) and an increase in $\delta^{13}$C values of our symbiont–bearing species if calcification temperature would dominate the $\delta^{13}$C signal. Since *G. ruber* w $\delta^{13}$C also showed a negative anomaly, we do not see the need to consider changes in calcification temperatures as a relevant driver for our $\delta^{13}$C anomalies. We briefly refer to this issue in the revised version of our manuscript.

Regarding the possible effect of changes in carbonate ion concentration, we are aware that changes in the seawater $[CO_3^{2-}]$ can impact planktonic foraminiferal $\delta^{13}$C in an inverse way (i.e., higher (lower) $[CO_3^{2-}]$ decreases (increases) planktonic foraminiferal $\delta^{13}$C) (Spero et al., 1997). Given the lack of regional upper ocean reconstructions for $[CO_3^{2-}]$, we assumed that increased $CO_{2atm}$ would be accompanied by a decrease in sea surface $[CO_3^{2-}]$ (Broecker and Peng, 1993). This would promote an increase in the $\delta^{13}C_{DIC}$. HS are frequently associated with an increase in $CO_{2atm}$ (Ahn and Brook, 2008; Ahn and Brook, 2014) but our records show a negative $\delta^{13}$C anomaly, suggesting that changes in $[CO_3^{2-}]$ are not the dominant driver of our $\delta^{13}$C anomalies. We briefly refer to this issue in the revised version of our manuscript.

Moreover, the authors suggest that increased runoff from the Plata river drainage basin led to increased sediment rate. It is an interesting result, which nicely fits with a southward shift of the ITCZ during that time, however river runoff might potentially have a fairly low d13C signature, thus potentially also influencing surface d13C at the core location?

Comment #5 - Abrupt millennial–scale climate events of the last glacial period have been associated with increased precipitation over tropical and subtropical South America to the east of the Andes (e.g., Arz et al., 1998; Wang et al., 2007; Kanner et al., 2012; Stríkis et al., 2015). During HS1, however, the millennial–scale signal of $\delta^{18}O_{IVC-sw}$ from the upper water column of our core site indicates an increase in salinity (Chiessi et al., 2015). Thus, the upper water column of our core site was not affected by an increase in freshwater discharge from the Plata River at millennial–scale. Since the anomaly of precipitation during HS1 was stronger than during HS3 and HS2 in the Plata River drainage basin (Wang et al., 2007), it is unlikely that weaker precipitation anomalies in the Plata River drainage basin (i.e., HS3 and HS2) affected the upper water column of our core site more intensely than during HS1. This suggests that changes in the discharge of the Plata River drainage basin at millennial–scale are not a relevant driver of our $\delta^{13}C$ anomalies.

Please note that: 1) the ice core data (Ahn and Brook 2014) do not support any atmospheric CO2 increase during HS2 and HS3.

Comment #6 - We cannot agree with this point. The best resolved $CO_{2atm}$ record available (Ahn and Brook, 2014) shows an increase in $CO_2$ during HS3 and HS2 that is highlighted by red arrows in the original publication (please see Fig. 1c from Ahn and Brook (2014)). The slight offset between the increase in $CO_{2atm}$ and the HS3 and HS2 intervals used in our manuscript can be accommodated by age model uncertainties.

2) D13CO2 can't really be used due to poor resolution and most likely issues with age model.

Comment #7 - We agree that the resolution of the $\delta^{13}CO_{2atm}$ curve is still suboptimal to resolve changes during HS2, and we have removed this curve from Fig. 5 of the revised version of our manuscript. We are aware of Eggleston et al.'s (2016) new $\delta^{13}CO_{2atm}$ data. However, despite their reference to a reduction in $\delta^{13}CO_{2atm}$ during HS2, their record also lacks the necessary temporal resolution to allow an appropriate comparison to our records.

3) The link between changes in southern hemispheric westerlies and AMOC changes is still poorly documented.

Comment #8 - Despite the existence of some open issues, several publications (e.g., Lee et al., 2003; Anderson et al., 2009; Toggweiler et al., 2006; Tschumi et al., 2008; d'Orgeville et al., 2010; Lourantou et al., 2010; Sigman et al., 2010; Tschumi et al.,

2011; Lee et al., 2011; Voigt et al., 2015) describe links between changes in AMOC and the Southern Hemisphere westerlies.

4) the opal flux in the Southern Ocean (Anderson et al. 2009) does not increase during HS2 and HS3.

Comment #9 - We are aware that Anderson et al.'s (2009) opal flux data (a proxy for the strength of Southern Ocean upwelling) did not change during the HS3 and HS2. However, this could be a temporal resolution issue. In core TN057-13-4PC, opal measurements show a mean temporal resolution of ca. 310 yr for HS1, but ca. 1550 yr for HS2 (this core does not reach HS3). In core E27-23, the mean temporal resolution of HS1 measurements is ca. 686 yr, but ca. 1550 yr for HS2, and ca. 1850 yr for HS3. HS1 opal measurements in core NBP9802-6PC have a mean temporal resolution of ca. 1715 yr, while HS2 measurements have a mean temporal resolution of ca. 3100 yr (this core does not reach HS3). Additionally, Anderson et al. (2009) suggest that such mechanism may be a common feature for other HS of the last glacial period.

As such the whole discussion section, as well as conclusion and abstract need to be rewritten.

Comment #10 - To account for the main topics raised by Referee #1 (but also Referee #2 and Dr. Schmittner) we have revised the necessary sections of our manuscript accordingly (please see above).

References

Ahn, J. and Brook, E. J.: Atmospheric $CO_2$ and climate on millennial time scales during the last glacial period, Science, 322, 83-85, doi:10.1126/science.1160832, 2008.

Ahn, J. and Brook, E. J.: Siple Dome ice reveals two modes of millennial $CO_2$ change during the last ice age, Nature, 5, doi:10.1038/ncomms4723, 2014.

Anderson, R. F., Ali, S., Bradtmiller, L. I., Nielsen, S. H. H., Fleisher, M. Q., Anderson, B. E., and Burckle, L. H.: Wind-Driven Upwelling in the Southern Ocean and the Deglacial Rise in Atmospheric $CO_2$, Science, 323, 1443-1448, doi:10.1126/science.1167441, 2009.

Arz, H. W., Pätzold, J., and Wefer, G.: Correlated millennial-scale changes in surface hydrography and terrigenous sediment yield inferred from last-glacial marine deposits off northeastern Brazil, Quaternary Res, 50, 157-166, doi:10.1006/qres.1998.1992, 1998.

Barker, S., Diz, P., Vautravers, M. J., Pike, J., Knorr, G., Hall, I. R., and Broecker, W. S.: Interhemispheric Atlantic seesaw response during the last deglaciation, Nature, 457, 1097-1102, doi:10.1038/nature07770, 2009.

Bemis, B. E., Spero, H. J., Lea, D. W., and Bijma, J.: Temperature influence on the carbon isotopic composition of *Globigerina bulloides* and *Orbulina universa* (planktonic foraminifera), Mar Micropaleontol, 38, 213–228, 2000.

Broecker, W. S., and Peng, T.-H.: What caused the glacial to interglacial $CO_2$ change? In The Global Carbon Cycle (ed. Heimann, M.) 95–115, Springer, Berlin, 1993.

Burke, A., and Robinson, L. F.: The Southern Ocean's role in carbon exchange during the last deglaciation, Science, 335, 557–561, 2012.

Charles, C. D., and Fairbanks, R. G.: Glacial to interglacial changed in the isotopic gradients of Southern Ocean surface water, in Geological History of the Polar Oceans: Arctic Versus Antarctic, edited by U. Bleil and J. Thiede, pp. 519-538, Kluwer Academic, Norwell, Mass., 1990.

Chiessi, C. M., Mulitza, S., Mollenhauer, G., Silva, J. B., Groeneveld, J., and Prange, M.: Thermal evolution of the western South Atlantic and the adjacent continent during Termination 1, Clim Past, 11, 915-929, doi:10.5194/cp-11-915-2015, 2015.

d'Orgeville, M., Sijp, W. P., England, M. H., and Meissner, K. J.: On the control of glacial-interglacial atmospheric $CO_2$ variations by the Southern Hemisphere westerlies, Geophys Res Lett, 37, L21703, 2010.

Eggleston, S., Schmitt, J., Bereiter, B., Schneider, R., and Fischer, H.: Evolution of the stable carbon isotope composition of atmospheric $CO_2$ over the last glacial cycle, Paleoceanography, 10.1002/2015PA002874, 2016.

Hemleben, C., Spindler, M., and Anderson, O. R.: Modern planktonic foraminifera (p. 363).New York: Springer, 1989.

Kanner, L. C., Burns, S. J., Cheng, H., and Edwards, R. L.: High-Latitude Forcing of the South American Summer Monsoon During the Last Glacial, Science, 335, 570-573, doi:10.1126/science.1213397, 2012.

Lee, S., and H. Kim: The dynamical relationship between subtropical and eddy-driven jets, J Atmos Sci, 60, 1490–1503, 2003.

Lee, S.-Y., Chiang, J. C. H., Matsumoto, K., and Tokos, K. S.: Southern Ocean wind response to North Atlantic cooling and the rise in atmospheric $CO_2$: Modeling perspective and paleoceanographic implications, Paleoceanography, 26, PA1214, doi:10.1029/2010PA002004, 2011.

Lourantou, A., Lavric, J. V., Kohler, P., Barnola, J. M., Paillard, D., Michel, E., Raynaud, D., and Chappellaz, J.: Constraint of the $CO_2$ rise by new atmospheric carbon isotopic measurements during the last deglaciation, Global Biogeochem Cy, 24, GB2015, doi:10.1029/2009gb003545, 2010.

Lynch-Stieglitz, J., Stocker, T. F., Broecker, W. S., and Fairbanks, R. G.: The influence of air-sea exchange on the isotopic composition of oceanic carbon: Observations and modeling, Global Biogeochem Cy, 9 (4), 653-665, 1995.

Menviel, L., Timmermann, A., Mouchet, A., and Timm, O.: Meridional reorganizations of marine and terrestrial productivity during Heinrich events, Paleoceanography, 23, PA1203, doi:10.1029/2007PA001445, 2008.

Oppo, D. W., and Fairbanks, R. G.: Carbon isotope composition of tropical surface water during the past 22,000 years, Paleoceanography, 4, 333-351, doi:10.1029/PA004i004p00333, 1989.

Schmittner, A., and Galbraith, E. D.: Glacial greenhouse-gas fluctuations controlled by ocean circulation changes, Nature, 456, doi:10.1038/nature07531, 2008.

Schmittner, A., and Lund, D. C.: Early deglacial Atlantic overturning decline and its role in atmospheric $CO_2$ rise inferred from carbon isotopes ($\delta^{13}C$), Clim Past, 11, 135–152, doi:10.5194/cp-11-135-2015, 2015.

Sigman, D. M., Hain, M. P., and Haug, G. H.: The polar ocean and glacial cycles in atmospheric $CO_2$ concentration, Nature, 466, 47–55, 2010.

Spero, H. J., and Lea, D. W.: Intraspecific stable isotope variability in the planktic foraminifera *Globigerinoides sacculifer*: Results from laboratory experiments, Mar Micropaleontol, 22, 221–234, 1993.

Spero, H. J., and Lea, D. W.: Experimental determination of stable isotope variability in *Globigerina bulloides*: Implications for paleoceanographic reconstruction, Mar Micropaleontol, 28, 231–246, 1996.

Spero, H. J., Bijma, J., Lea, D., and Bemis, B. E.: Effect of seawater carbonate concentration on foraminiferal carbon and oxygen isotopes, Nature, 390, 1997.

Stríkis, N. M., Chiessi, C. M., Cruz, F. W., Vuille, M., Cheng, H., de Souza Barreto, E. A., Mollenhauer, G., Kasten, S., Karmann, I., Edwards, R. L., Pablo Bernal, J., and Sales, H. D. R.: Timing and structure of Mega-SACZ events during Heinrich Stadial 1, Geophys Res Lett, 42, 5477-5484, doi:10.1002/2015GL064048, 2015.

Toggweiler, J. R., Russell, J. L., and Carson, S. R.: Midlatitude westerlies, atmospheric $CO_2$, and climate change during the ice ages, Paleoceanography, 21, PA2005, 2006.

Tschumi, T., Joos, F., and Parekh, P.: How important are Southern Hemisphere wind changes for low glacial carbon dioxide? A model study, Paleoceanography, 23, PA4208, 2008.

Tschumi, T., Joos, F., Gehlen, M., and Heinze, C.: Deep ocean ventilation, carbon isotopes, marine sedimentation and the deglacial $CO_2$ rise, Clim Past, 7, 771–800, doi:10.5194/cp-7-771-2011, 2011.

Voigt, I., Chiessi, C. M., Prange, M., Mulitza, S., Groeneveld, J., Varma, V., and Henrich, R.: Holocene shifts of the southern westerlies across the South Atlantic, Paleoceanography, 10.1002/2014PA002677, 2015.

Wang, X., Auler, A. S., Edwards, R. L., Cheng, H., Ito, E., Wang, Y., Kong, X., and Solheid, M.: Millennial-scale precipitation changes in southern Brazil over the past 90,000 years, Geophys Res Lett, 34, doi:10.1029/2007GL031149, 2007.

---

## Author Comment (AC3) · 14 Oct 2016

We thank Referee #2 for the constructive review of our manuscript. To facilitate the discussion, we have copied his/her comments below in black and inserted our responses in green.

Review of "Glacial d13C decreases in the western South Atlantic forced by millennial changes in Southern Ocean ventilation" by Campos et al.

The authors use the d13C of planktonic foraminifera to infer variability in the d13C of DIC in surface/thermocline waters in the SW Atlantic. One strength of the paper is the comprehensive review of the modern oceanographic setting. Another is that the planktonic time series are based on a sediment core with very high sedimentation rates, allowing for high resolution reconstruction of surface ocean d13C during Heinrich Stadial 2 and 3. It is also important that the authors used two different species to reconstruct d13C (one surface dwelling and another thermocline dwelling) to account for potential biases in habitat and vital effects that could overprint changes in the d13C of DIC. Given the high resolution and replicated nature of the record, the authors clearly show that this part of the Southwest Atlantic underwent significant changes in d13C during Heinrich Stadial 2 and 3.

The primary weakness of the paper is the interpretation of the d13C and sedimentation rate results. The authors are quick to assume that their d13C records reflect the input of light carbon from the Southern Ocean and neglect other possible explanations that are well documented in the published literature.

Comment #1 - We agree that other mechanisms could have also contributed to our records. Considering this comment and similar suggestions from Referee #1 and Dr. Schmittner, we have incorporated two additional hypotheses (see below) in the revised version of our manuscript, in order to have a more thorough and balanced discussion:

  i.    AMOC–induced weakening of the biological pump

As shown by model experiments (e.g., Schmittner and Galbraith, 2008; Schmittner and Lund, 2015), AMOC slowdown events may cause a decrease in the global efficiency of the oceanic biological pump, being an important driver for the oceanic $CO_2$ outgassing during the last deglaciation, i.e., HS1 (but most likely also during other HS, e.g., HS3 and HS2). Two factors acting in conjunction are claimed to be responsible for the simulated weakening of the efficiency of the biological pump (Schmittner and Galbraith, 2008). First, the reduction in North Atlantic Deep Water (NADW) formation decreases the input of low preformed nutrient (high $\delta^{13}C_{DIC}$ values) waters to the interior ocean which becomes more dominated by the high preformed nutrient southern component (low $\delta^{13}C_{DIC}$ values) waters (e.g., Antarctic Bottom Water and Antarctic Intermediate Water). Second, the reduction of the Southern Ocean stratification induced by the decrease of salt input via NADW formation promotes the strengthening of the upwelling and subsequent sinking of high preformed nutrient (low $\delta^{13}C_{DIC}$ values) waters to the interior ocean, thus reducing the capacity of those unutilized nutrients to sequester carbon via the biological pump.

The weakening of the biological pump would promote the accumulation of isotopically light organic carbon in the upper water column of the South Atlantic and the negative anomaly of the $\delta^{13}C_{DIC}$ as well as of the $\delta^{13}CO_{2atm}$ (Schmittner and Lund, 2015). This $\delta^{13}C_{DIC}$ anomaly should be captured by the tests of planktonic foraminifera $\delta^{13}C$ because they incorporate $\delta^{13}C_{DIC}$ during calcification (Spero and Lea, 1996; Bemis et al., 2000). Our planktonic foraminifera $\delta^{13}C$ records agree with this hypothesis by showing negative anomalies during HS3 and HS2. Additionally, this mechanism also provides a possible explanation for the larger negative $\delta^{13}C$ anomaly in *G. ruber* w (mixed layer–dwelling) relative to the anomaly in *G. inflata* (permanent thermocline–dwelling).

ii.     the role of air–sea gas exchange

In the original version of our manuscript (page 7, line 13) we briefly mentioned the role of air–sea gas exchange. However, we see the need to further expand this topic in order to have a more thorough and balanced discussion.

We are aware that the $\delta^{13}C_{DIC}$ of the surface ocean can be affected by air–sea gas exchange (Oppo and Fairbanks, 1989; Charles and Fairbanks, 1990; Lynch-Stieglitz et al., 1995). Therefore, we cannot exclude the possibility that the likely decrease in $\delta^{13}CO_{2atm}$ during AMOC slowdown events (Eggleston et al., 2016) (e.g., HS3 and HS2) could have affected the $\delta^{13}CO_{2DIC}$ via air–sea gas exchange, especially in regions of water mass formation. The formation regions of South Atlantic Central Water and Tropical Water are areas of major ocean $CO_2$ uptake and may contribute to the $\delta^{13}C$ anomalies observed in our records. Additionally, since the isotopic fractionation during air–sea exchange is temperature–dependent, the weakening of the AMOC and subsequent warming of the upper subtropical South Atlantic (Barker et al. 2009; Chiessi et al. 2015) could also have contributed to the observed $\delta^{13}C$ anomalies in our records (as suggested by Referee #2).

However, for the reasons stated below we still consider the possible role played by changes in Southern Ocean ventilation in producing our $\delta^{13}C$ anomalies:

i.     the mechanisms that invoke the Southern Ocean as one of the $CO_{2atm}$–sources during the last deglaciation are based on reconstructions as well as conceptual and numerical models that, to our understanding, have not yet been negated (Toggweiler et al., 2006; Anderson et al., 2009; Sigman et al., 2010; Tschumi et al., 2011; Lee et al., 2011; Burke and Robinson, 2012);

ii.     model experiments that question the role of the Southern Ocean (e.g., Schmittner and Lund, 2015) were performed under preindustrial boundary conditions, which can significantly influence the results (e.g., Menviel et al., 2008), and do not constitute a firm negation of the Southern Ocean hypothesis.

The authors also invoke speculative connections between rainfall and sedimentation rate in the core as support for their climate interpretation, despite disagreement between the sedimentation rate and planktonic d13C patterns.

Comment #2 - We agree that the second HS2 peak in sedimentation rate and the respective planktonic $\delta^{13}$C anomaly are not perfectly aligned in time. However, this apparent offset can be exclusively due to (i) the occurrence of $^{14}$C plateaus during HS (e.g., Sarnthein et al., 2007; Franke et al., 2008), and/or (ii) the discretized way our age model was produced in relation to the "continuous" $\delta^{13}$C measurements.

Finally, the authors neglect to mention much of the work at the Brazil Margin spanning the last deglaciation (including Heinrich Stadial 1) that is both relevant to their work and inconsistent with a Southern Ocean driver.

Comment #3 - We have included relevant publications from the Brazil Margin (e.g., Tessin and Lund, 2013; Lund et al. 2015) in the revised version of our manuscript together with a thorough and more balanced discussion (please see our Comment #1 above). However, the hypothesis that the Southern Ocean acts as a possible $CO_{2atm}$– source during HS has, to the best of our knowledge, not yet been proven incorrect and we still see the need to discuss it together with other possible explanations (please see our Comment #1 above).

Major issues:

While the prevailing view is that Southern Ocean outgassing drove surface ocean d13C anomalies during the last deglaciation, the authors also need to reference to Andreas Schmittner's work that shows that weakening of the AMOC can alter the preformed nutrient budget of the global ocean and therefore the efficiency of the biological pump. Weakening of the biopump would preferentially leave light carbon in the surface and create negative d13C anomalies in multiple ocean basins (e.g. Schmittner, 2005; Schmittner and Galbraith, 2008).

Also, detailed reconstructions from the Brazil Margin show that benthic d13C and d18O changed late in the deglaciation, suggesting that the abyssal ocean was an unlikely source for the surface ocean anomalies during HS1 (Lund et al., 2015). Additional published records from the SE Atlantic shows a similar pattern, with a late deglacial response in the abyssal records (Waelbroeck et al., 2011; Roberts et al., 2016).

Comment #4 – Considering that the role of the weakening of the biological pump was also raised by Referee #1 and Dr. Schmittner, we have incorporated this hypothesis in the revised version of our manuscript (please see our Comment #1 above).

Our records complement the Brazil Margin depth transect (e.g., Tessin and Lund, 2013; Lund et al., 2015) with upper water column high temporal resolution data that also agree with the model output of Schmittner and Lund (2015) by suggesting a decrease in $\delta^{13}C_{DIC}$ at the upper water column of the western South Atlantic.

However, it is worth noting that while Tessin and Lund (2013) and Lund et al. (2015), for example, focus on bottom waters, we concentrate on the upper water column. In the revised version of our manuscript, we suggest that three processes may have contributed to the negative $\delta^{13}C$ anomalies in our records during HS3 and HS2: (i) changes in Southern Ocean ventilation; (ii) AMOC–induced weakening of the biological pump; and (iii) changes in air–sea gas exchange.

Furthermore, intermediate depth d13C reconstructions from the Brazil Margin (Oppo and Horowitz, 2000; Curry and Oppo, 2005; Lund et al., 2015) imply that AAIW and SAMW were not carriers of a light carbon signal during HS1, which is inconsistent with the mechanism invoked by Campos et al. While the authors did a commendable job summarizing the literature pertaining the modern oceanographic setting, the introductory material includes surprisingly little information from previously published work along the Brazil Margin. As a result, the introductory material is incomplete and the lack of context limits the interpretation in the discussion section.

Comment #5 - To account to this additional mechanism in the revised manuscript we have inserted a new section describing the AMOC–induced weakening of the biological pump (please see our Comment #1 above).

The other main issue with the data interpretation is that the continental hydroclimate response isn't well supported by the sedimentation rate data. While sedimentation rates peak early in HS2, at about the same time as a planktonic d13C minimum, there is a second peak in sedimentation rate after HS2 that corresponds to a broad maximum in d13C. Furthermore, there is no peak in sedimentation rate during either HS3 or HS1. Taken as a whole, the sedimentation rate data therefore do not support a continental hydroclimate connection, which perhaps isn't surprising given the many factors that can influence sedimentation rates at a single core site.

Comment #6 - Please see our Comment #2 above. Additionally, although HS3 seems not to be related to a positive anomaly in sedimentation rate at our core site, HS1 and the Younger Dryas are both related to positive anomalies (of smaller amplitude during the Younger Dryas if compared to the anomalies during HS2 and HS1). The relatively high sea–level during HS3 and the Younger Dryas if compared to the intervening period (Waelbroeck et al., 2002) may have dampened a more significant anomaly in sedimentation rates during HS3 and the Younger Dryas (e.g., Lantzsch et al., 2014). This discussion has not been inserted in the revised manuscript because we have focused in HS3 and HS2.

Finally, the manuscript would benefit from inclusion of the planktonic d18O data. If there is a clear hydroclimate signal at this site, it could appear in the d18O data as intervals of unusually low d18O. The d18O data would also be informative for assessing the influence of air-sea gas exchange effects on the d13C signal. The d13C of surface ocean DIC is influenced by temperature dependent gas exchange, with a relationship of -0.1 per mil per deg C of warming (Broecker and Maier-Reimer, 1992; Lynch-Stieglitz et al., 1995). While such an effect is unlikely to explain the full

magnitude of the d13C signals, it should be included in the discussion. Model results suggest that weakening of the AMOC warms the upper subtropical South Atlantic by 2-3 deg C (Marcotte et al., 2011) which would account for up to 0.3 per mil of the planktonic d13C anomalies.

Comment #7 - A similar topic was also raised by Referee #1. Please refer to our Comment #5 to Referee #1. Additionally, a section expanding the discussion on the possible role of air–sea gas exchange is included in the revised version of our manuscript. This mechanism is described in our Comment #1 above.

It is worthy of note that our foraminiferal $\delta^{18}$O data do not present clear trends during HS3 and HS2. The $\delta^{18}$O of planktonic foraminifera is markedly influenced by the temperature and salinity of the sea water where the tests calcified (e.g., Rohling and Cooke, 1999). The relationship between temperature and $\delta^{18}$O of planktonic foraminifera is inversely proportional, and the relationship between salinity and $\delta^{18}$O of planktonic foraminifera is directly proportional. Therefore, higher temperatures decrease foraminiferal $\delta^{18}$O, whereas higher salinities increase foraminiferal $\delta^{18}$O (e.g., Rohling and Cooke, 1999).

For periods and regions of the global ocean where an increase (decrease) in temperature is associated to a decrease (increase) in salinity, the $\delta^{18}$O of planktonic foraminifera will clearly record such environmental changes (e.g., Guilderson and Pak, 2007). However, for those cases where temperature and salinity increase (decrease) simultaneously, these environmental changes may not be clearly recorded in foraminiferal $\delta^{18}$O. The last scenario seems to be the case for the upper water column under the influence of the Brazil Current for centennial-scale events of the Holocene (Chiessi et al., 2014), as well as for HS1 (Chiessi et al., 2015), and possibly also for HS3 and HS2 (given the lack of regional SST reconstructions for HS3 and HS2).

Minor issues:

Page 6, Line 12: Here the authors should consider the possibility that weakening of the biological pump could have produced d13C anomalies not only locally, but on a larger spatial scales.

Comment #8 - Agreed. We have changed the revised version of our manuscript accordingly.

Page 6, Line 20: It is important to mention that published mode and intermediate water records from the Brazil Margin either show a positive d13C anomaly during HS1 or a delayed negative d13C response that is inconsistent the light carbon being transported northward via mode and intermediate water.

Comment #9 - Agreed (please see our Comment #1 above). We have revised the manuscript accordingly. Additionally, we also incorporate two more hypotheses (i.e., (i) AMOC–induced weakening of the biological pump, and (ii) the role of air–sea gas

exchange) to the revised version of our manuscript, in order to have a more thorough and balanced discussion.

Page 7, Line 5: The authors seem to take it as a given that Southern Hemisphere westerlies drives greater upwelling of deep waters when it actuality there is limited data to support such a link. Please qualify these types of statements to reflect the limited constraints that exist.

Comment #10 – Agreed (please see our Comment #8 to Referee #1). We have revised our manuscript accordingly.

Page 7, Line 9: See comments above that published intermediate depth records from the Brazil Margin don't show clear negative carbon isotope anomalies during HS1, which is relevant to whether a similar process occurred during HS2 and HS3.

Comment #11 - Agreed (please see our Comment #1 above). We have revised our manuscript accordingly. Additionally we also incorporate two more hypotheses (i.e., (i) AMOC–induced weakening of the biological pump, and (ii) the role of air–sea gas exchange) to the revised version of our manuscript, in order to have a more thorough and balanced discussion.

Figure 4: If the authors are going to pursue the sedimentation rate-hydoclimate link, then the sedimentation rates should also be included in this figure for comparison to the d13C. The d18O records should also be included to assess the potential impact of sea surface temperature on d13C.

Comment #12 - Agreed. We have inserted the sedimentation rate record in Figure 5 of the revised version of our manuscript. Regarding the $\delta^{18}$O data, please refer to our Comment #5 to Referee #1 and our Comment #7 above.

Figure 5: There are several time series in this plot that aren't essential for the discussion, such as the Iberian Margin SST record, the EDML d18O time series, and the Siple Dome CO2.

Comment #13 - We agree that removing the $\delta^{18}$O EDML curve (EPICA Community Members, 2006) would not harm the main focus of our manuscript. In addition, we have also removed the Taylor Dome $\delta^{13}CO_{2atm}$ curve (Smith et al., 1999) for the reason mentioned in our Comment #7 to Referee #1.

However, we would prefer not to delete the following curves: (i) the Iberian Margin SST curve (Bard, 2002) because it shows a "w–structure" during HS2 that gives support to the proposed "w–structure" of HS2; and (ii) the Siple Dome $CO_{2atm}$ (Ahn and Brook, 2014) due to its fundamental importance to our manuscript (please see our Comment #6 to Referee #1).

References

Ahn, J. and Brook, E. J.: Siple Dome ice reveals two modes of millennial $CO_2$ change during the last ice age, Nature, 5, doi:10.1038/ncomms4723, 2014.

Anderson, R. F., Ali, S., Bradtmiller, L. I., Nielsen, S. H. H., Fleisher, M. Q., Anderson, B. E., and Burckle, L. H.: Wind-Driven Upwelling in the Southern Ocean and the Deglacial Rise in Atmospheric $CO_2$, Science, 323, 1443-1448, doi:10.1126/science.1167441, 2009.

Bard, E.: Climate shock: Abrupt changes over millennial time scales, Phys Today, 55, 32-38, 2002.

Barker, S., Diz, P., Vautravers, M. J., Pike, J., Knorr, G., Hall, I. R., and Broecker, W. S.: Interhemispheric Atlantic seesaw response during the last deglaciation, Nature, 457, 1097-1102, doi:10.1038/nature07770, 2009.

Bemis, B. E., Spero, H. J., Lea, D. W., and Bijma, J.: Temperature influence on the carbon isotopic composition of *Globigerina bulloides* and *Orbulina universa* (planktonic foraminifera), Mar Micropaleontol, 38, 213–228, 2000.

Burke, A., and Robinson, L. F.: The Southern Ocean's role in carbon exchange during the last deglaciation, Science, 335, 557–561, 2012.

Charles, C. D., and Fairbanks, R. G.: Glacial to interglacial changed in the isotopic gradients of Southern Ocean surface water, in Geological History of the Polar Oceans: Arctic Versus Antarctic, edited by U. Bleil and J. Thiede, pp. 519-538, Kluwer Academic, Norwell, Mass., 1990.

Chiessi, C. M., Mulitza, S., Groeneveld, J., Silva, J. B., Campos, M. C., and Gurgel, M. H. C.: Variability of the Brazil Current during the late Holocene, Palaeogeogr Palaeocl, 415, 28-36, 2014.

Chiessi, C. M., Mulitza, S., Mollenhauer, G., Silva, J. B., Groeneveld, J., and Prange, M.: Thermal evolution of the western South Atlantic and the adjacent continent during Termination 1, Clim Past, 11, 915-929, doi:10.5194/cp-11-915-2015, 2015.

Eggleston, S., Schmitt, J., Bereiter, B., Schneider, R., and Fischer, H.: Evolution of the stable carbon isotope composition of atmospheric CO2 over the last glacial cycle, Paleoceanography, 10.1002/2015PA002874, 2016.

EPICA Community Members: One-to-one coupling of glacial climate variability in Greenland and Antarctica, Nature, 444, 195-198, doi:10.1038/nature05301, 2006.

Franke, J., Paul, A., and Schulz, M.: Modeling variations of marine reservoir ages during the last 45 000 years, Clim Past, 4, 2, 125-136, doi:10.5194/cp-4-125-2008, 2008.

Guilderson, T. P., and Pak, D. K.: Salinity proxies $\delta^{18}O$. In: Encyclopedia Of Quaternary: Paleoceanography, Physical and Chemical proxies. Elsevier: Amsterdam, p. 1766-1775, 2007.

Lantzsch, H., Hanebuth, T. J. J., Chiessi, C. M., Schwenk, T., and Violante, R. A.: The high-supply, current-dominated continental margin of southeastern South America during the late Quaternary, Quaternary Res, 81, 339-354, doi:10.1016/j.yqres.2014.01.003, 2014.

Lee, S.-Y., Chiang, J. C. H., Matsumoto, K., and Tokos, K. S.: Southern Ocean wind response to North Atlantic cooling and the rise in atmospheric $CO_2$: Modeling perspective and paleoceanographic implications, Paleoceanography, 26, PA1214, doi:10.1029/2010PA002004, 2011.

Lund, D. C., Tessin, A. C., Hoffman, J. L., and Schmittner, A.: Southwest Atlantic water mass evolution during the last deglaciation, Paleoceanography, 10.1002/2014PA002657, 2015.

Lynch-Stieglitz, J., Stocker, T. F., Broecker, W. S., and Fairbanks, R. G.: The influence of air-sea exchange on the isotopic composition of oceanic carbon: Observations and modeling, Global Biogeochem Cy, 9 (4), 653-665, 1995.

Menviel, L., Timmermann, A., Mouchet, A., and Timm, O.: Meridional reorganizations of marine and terrestrial productivity during Heinrich events, Paleoceanography, 23, PA1203, doi:10.1029/2007PA001445, 2008.

Oppo, D. W., and Fairbanks, R. G.: Carbon isotope composition of tropical surface water during the past 22,000 years, Paleoceanography, 4, 333-351, doi:10.1029/PA004i004p00333, 1989.

Rohling, E. J., and Cooke, E.: Stable oxygen and carbon isotopes in foraminiferal carbonate shells. In: Sen Gupta, B. K.,: Modern Foraminifera. Kluwer Academic Publishers, p. 239-259, 1999.

Sarnthein, M., Grootes, P. M., Kennett, J. P., and Nadeau, M.-J.: $^{14}C$ reservoir ages show deglacial changes in ocean currents and carbon cycle. In: Ocean Circulation: Mechanisms and Impacts - Past and Future Changes of Meridional Overturning (eds A. Schmittner, J. C. H. Chiang and S. R. Hemming), American Geophysical Union, Washington, D. C.. doi: 10.1029/173GM13, 2007.

Schmittner, A., and Galbraith, E. D.: Glacial greenhouse-gas fluctuations controlled by ocean circulation changes, Nature, 456, doi:10.1038/nature07531, 2008.

Schmittner, A., and Lund, D. C.: Early deglacial Atlantic overturning decline and its role in atmospheric $CO_2$ rise inferred from carbon isotopes ($\delta^{13}C$), Clim Past, 11, 135–152, doi:10.5194/cp-11-135-2015, 2015.

Sigman, D. M., Hain, M. P., and Haug, G. H.: The polar ocean and glacial cycles in atmospheric $CO_2$ concentration, Nature, 466, 47–55, 2010.

Smith, H. J., Fischer, H., Wahlen, M., Mastroianni, D., and Deck, B.: Dual modes of the carbon cycle since the Last Glacial Maximum, Nature, 400, 248-250, doi:10.1038/22291, 1999.

Spero, H. J., and Lea, D. W.: Experimental determination of stable isotope variability in *Globigerina bulloides*: Implications for paleoceanographic reconstruction, Mar Micropaleontol, 28, 231–246, 1996.

Tessin, A. C., and Lund, D. C.: Isotopically depleted carbon in the mid-depth South Atlantic during the last deglaciation, Paleoceanography, 28, 296–306, 2013.

Toggweiler, J. R., Russell, J. L., and Carson, S. R.: Midlatitude westerlies, atmospheric $CO_2$, and climate change during the ice ages, Paleoceanography, 21, PA2005, 2006.

Tschumi, T., Joos, F., Gehlen, M., and Heinze, C.: Deep ocean ventilation, carbon isotopes, marine sedimentation and the deglacial $CO_2$ rise, Clim Past, 7, 771–800, doi:10.5194/cp-7-771-2011, 2011.

Waelbroeck, C., Labeyrie, L., Michel, E., Duplessy, J. C., McManus, J. F., Lambeck, K., Balbon, E., and Labracherie. M.: Sea-level and deep water temperature changes derived from benthic foraminifera isotopic records, Quaternary Sci Rev, 21, 295-305, 2002.

---

## Author Response (AR1)

**1. Point-by-point response and relevant changes performed in the manuscript**

In this section we inserted a point-by-point response to the Referees as well as Dr. Schmittner together with all relevant changes performed in the manuscript. To facilitate the review, we copied the Referees´ as well as Dr. Schmittner´s comments below in black and inserted our comments in green. The modifications performed to the manuscript are shown in italic green in quotes.

**1.1 Responses to Referee #1**

We thank Referee #1 for the constructive review of our manuscript.

This is an interesting manuscript presenting two high-resolution planktic d13C records from the Brazilean coast covering HS3 and HS2. It is worth publishing in Climate of the Past if the "Discussion" section is completely rewritten and therefore the interpretation of the results re-assessed.

The authors have not shown that the planktic d13C decrease measured in their core was due to stronger Southern Ocean upwelling. This is just an hypothesis.

Comment #1 - We agree that it is a hypothesis and have now treated it accordingly by rephrasing specific parts of the revised version of our manuscript.

Additionally considering this comment and the similar suggestions from Referee #2 and Dr. Schmittner, we also incorporated two additional hypotheses (see below) in the revised manuscript, in order to have a more thorough and balanced discussion:

[revised manuscript text omitted]

In the conclusions we also inserted the two additional hypotheses described above and removed the mention of the Southern Ocean as the main source of the $CO_2$ outgassed to the atmosphere during HS3 and HS2.

Page 13, lines 14-18: we added *"(ii) Weakening of the global oceanic biological pump. A weak AMOC during HS3 and HS2 would promote an accumulation of $^{13}C$–depleted $CO_2$ in the upper water column of the South Atlantic. This accumulation would result in a negative anomaly of the $\delta^{13}C_{DIC}$ (as well as of the $\delta^{13}CO_{2atm}$) that in turn would be captured by the tests of planktonic foraminifera at our core site. We further suggest that changes in air–sea gas exchange could have contributed to the decreases in $\delta^{13}C$ via both mechanisms.".*

Page 13, line 20: we removed *"of the Southern Ocean"*.

However, for the reasons stated below we still consider the possible role played by changes in Southern Ocean ventilation in producing our $\delta^{13}C$ anomalies:

i.   the mechanisms that invoke the Southern Ocean as one of the $CO_{2atm}$–sources during the last deglaciation are based on reconstructions as well as conceptual and numerical models that, to our understanding, have not yet been negated (Toggweiler et al., 2006; Anderson et al., 2009; Sigman et al., 2010; Tschumi et al., 2011; Lee et al., 2011; Burke and Robinson, 2012);

ii.     model experiments that question the role of the Southern Ocean (e.g., Schmittner and Lund, 2015) were performed under preindustrial boundary conditions, which can significantly influence the results (e.g., Menviel et al., 2008), and do not constitute a firm negation of the Southern Ocean hypothesis.

Planktic d13C is influenced by several factors such as changes in oceanic circulation, mixing, SST, export production: : : (see Charles et al. 1993, Lynch-Stieglitz et al. 1995, Menviel et al. 2015: : :).

Comment #2 - We agree that planktonic foraminiferal $\delta^{13}$C can be affected by several factors and that some of them where not thoroughly considered in the original version of our manuscript. To account for these different factors we have inserted two new sections (i.e., 5.2 Millennial–scale changes: AMOC–induced weakening of the biological pump, and 5.3 Millennial–scale changes: the role of air–sea gas exchange) in our manuscript (please see our Comment #1 above) and clearly refer to the complexity of the proxy.

Additionally, by analyzing two different species of planktonic foraminifera, i.e., *G. ruber* w that is a symbiont–bearing species, and *G. inflata* that is a symbiont–facultative species (e.g., Hemleben et al., 1989), we have excluded the potential bias in our records related to changing symbiont activity (e.g., Spero and Lea, 1993; Spero et al., 1997; Bemis et al., 2000). Since the $\delta^{13}$C record of both species show a similar pattern, changing symbiont activity can be disregarded as a potential bias to our results.

Page 4, lines 19-21: "*However, other factors such as calcification temperature, carbonate ion concentration, symbiont activity and air–sea gas exchange may also influence planktonic foraminiferal $\delta^{13}$C (Lynch-Stieglitz et al., 1995; Spero and Lea, 1996, Spero et al., 1997; Bemis et al., 2000).*".

Page 7, lines 17-26: "*During HS, we expect warmer temperatures to have occurred in the upper water column of the western South Atlantic (Barker et al. 2009; Chiessi et al. 2015). This would trigger an increase in $\delta^{13}$C values of the symbiont–bearing species investigated here if calcification temperature would dominate the $\delta^{13}$C signal (Bemis et al., 2000), which is not the case (Fig. 4a). Additionally, given the lack of regional upper ocean reconstructions for carbonate ion concentration, we assume that increased $CO_{2atm}$ that is frequently associated with HS (Ahn and Brook, 2008; Ahn and Brook,*

*2014) would have been accompanied by a decrease in sea surface carbonate ion concentration (Broecker and Peng, 1993). This would promote an increase in the $\delta^{13}C_{DIC}$ but our records show a negative $\delta^{13}C$ anomaly (Fig. 4). Furthermore, we analysed a symbiont–bearing and a facultative–symbiont species (i.e., G. ruber w and G. inflata, respectively) and both records show a similar pattern (Fig. 4) indicating that changes in symbiont activity can also be disregarded as a factor influencing our results (Spero et al., 1997; Bemis et al., 2000)."*

An AMOC decrease will lead to significant surface ocean d13C changes. Changes in Southern Ocean upwelling are not the only solution.

Comment #3 - Agreed. As mentioned in our Comment #1 (please see above) we have revised our manuscript accordingly by incorporating two additional mechanisms (i.e., 5.2 Millennial–scale changes: AMOC–induced weakening of the biological pump, and 5.3 Millennial–scale changes: the role of air–sea gas exchange) that could account for the $\delta^{13}C$ changes in our records.

In addition, it has been shown that during calcification temperature and carbonate ion content could have an impact on calcite d13C in planktic species (see Spero et al. 1997, Bemis et al. 2000: : :).

Comment #4 - Experiments with *O. universa* (a symbiont–bearing species) and *G. bulloides* (a non–symbiont species) suggest that under higher temperatures the first species presents higher $\delta^{13}C$ values whereas the second species presents lower $\delta^{13}C$ values (Bemis et al., 2000). These opposed $\delta^{13}C$ responses were related to the presence (i.e., *O. universa*) or absence (i.e., *G. bulloides*) of symbionts. We analyzed a symbiont–bearing and a facultative–symbiont species (i.e., *G. ruber* w and *G. inflata*, respectively), but both records showed a similar behavior. During HS, we would expect warmer temperatures in the upper water column of the western South Atlantic (Barker et al. 2009; Chiessi et al. 2015) and an increase in $\delta^{13}C$ values of our symbiont–bearing species if calcification temperature would dominate the $\delta^{13}C$ signal. Since *G. ruber* w $\delta^{13}C$ also showed a negative anomaly, we do not see the need to consider changes in calcification temperatures as a relevant driver for our $\delta^{13}C$ anomalies. We briefly refer to this issue in the revised version of our manuscript.

Page 7, lines 17-20: "*During HS, we expect warmer temperatures to have occurred in the upper water column of the western South Atlantic (Barker et al. 2009; Chiessi et al. 2015). This would trigger an increase in $\delta^{13}C$ values of the symbiont–bearing species investigated here if calcification temperature would dominate the $\delta^{13}C$ signal (Bemis et al., 2000), which is not the case (Fig. 4a).*".

Regarding the possible effect of changes in carbonate ion concentration, we are aware that changes in the seawater $[CO_3^{2-}]$ can impact planktonic foraminiferal $\delta^{13}C$ in an inverse way (i.e., higher (lower) $[CO_3^{2-}]$ decreases (increases) planktonic foraminiferal $\delta^{13}C$) (Spero et al., 1997). Given the lack of regional upper ocean reconstructions for $[CO_3^{2-}]$, we assumed that increased $CO_{2atm}$ would be accompanied by a decrease in sea surface $[CO_3^{2-}]$ (Broecker and Peng, 1993). This would promote an increase in the $\delta^{13}C_{DIC}$. HS are frequently associated with an increase in $CO_{2atm}$ (Ahn and Brook, 2008; Ahn and Brook, 2014) but our records show a negative $\delta^{13}C$ anomaly, suggesting that changes in $[CO_3^{2-}]$ are not the dominant driver of our $\delta^{13}C$ anomalies. We briefly refer to this issue in the revised version of our manuscript.

Page 7, lines 20-23: "*Additionally, given the lack of regional upper ocean reconstructions for carbonate ion concentration, we assume that increased $CO_{2atm}$ that is frequently associated with HS (Ahn and Brook, 2008; Ahn and Brook, 2014) would have been accompanied by a decrease in sea surface carbonate ion concentration (Broecker and Peng, 1993). This would promote an increase in the $\delta^{13}C_{DIC}$ but our records show a negative $\delta^{13}C$ anomaly (Fig. 4).*".

Moreover, the authors suggest that increased runoff from the Plata river drainage basin led to increased sediment rate. It is an interesting result, which nicely fits with a southward shift of the ITCZ during that time, however river runoff might potentially have a fairly low d13C signature, thus potentially also influencing surface d13C at the core location?

Comment #5 - Abrupt millennial–scale climate events of the last glacial period have been associated with increased precipitation over tropical and subtropical South America to the east of the Andes (e.g., Arz et al., 1998; Wang et al., 2007; Kanner et al., 2012; Stríkis et al., 2015). During HS1, however, the millennial–scale signal of $\delta^{18}O_{IVC-SW}$ from the upper water column of our core site indicates an increase in salinity

(Chiessi et al., 2015). Thus, the upper water column of our core site was not affected by an increase in freshwater discharge from the Plata River at millennial–scale. Since the anomaly of precipitation during HS1 was stronger than during HS3 and HS2 in the Plata River drainage basin (Wang et al., 2007), it is unlikely that weaker precipitation anomalies in the Plata River drainage basin (i.e., HS3 and HS2) affected the upper water column of our core site more intensely than during HS1. This suggests that changes in the discharge of the Plata River drainage basin at millennial–scale are not a relevant driver of our $\delta^{13}C$ anomalies.

Pages 12-13, lines 27-3: "*Some aspects of the regional response to HS1 are useful to evaluate this possibility. During HS1, ice volume corrected seawater–$\delta^{18}O$ from the upper water column of our core site indicates an increase in salinity (Chiessi et al., 2015). Thus, despite of the increased terrigenous discharge, it seems that the upper water column of our core site was not affected by an increase in freshwater discharge from the Plata River during HS1. Since the precipitation anomaly of HS1 was stronger than that of HS3 and HS2 in the Plata River drainage basin (Wang et al., 2007), it is unlikely that weaker precipitation anomalies of HS3 and HS2 would have impacted the upper water column of our core site more intensely than during HS1. This suggests that changes in the discharge of the Plata River drainage basin at millennial–scale are not a relevant driver of our $\delta^{13}C$ decreases, and that the buoyant low salinity waters were advected elsewhere by winds, while terrigenous sediments were already too deep to be influenced by the wind.*".

Please note that: 1) the ice core data (Ahn and Brook 2014) do not support any atmospheric CO2 increase during HS2 and HS3.

Comment #6 - We cannot agree with this point. The best resolved $CO_{2atm}$ record available (Ahn and Brook, 2014) shows an increase in $CO_2$ during HS3 and HS2 that is highlighted by red arrows in the original publication (please see Fig. 1c from Ahn and Brook (2014)). The slight offset between the increase in $CO_{2atm}$ and the HS3 and HS2 intervals used in our manuscript can be accommodated by age model uncertainties.

2) D13CO2 can't really be used due to poor resolution and most likely issues with age model.

Comment #7 - We agree that the resolution of the $\delta^{13}CO_{2atm}$ curve is still suboptimal to resolve changes during HS2, and we have removed this curve from Fig. 5 of the revised version of our manuscript. We are aware of Eggleston et al.'s (2016) new $\delta^{13}CO_{2atm}$ data. However, despite their reference to a reduction in $\delta^{13}CO_{2atm}$ during HS2, their record also lacks the necessary temporal resolution to allow an appropriate comparison to our records.

Page 9, lines 30-31: Despite the low temporal resolution, "*Eggleston et al.'s (2016) Antarctic $\delta^{13}CO_{2atm}$ record shows a decrease during HS2.*'

3) The link between changes in southern hemispheric westerlies and AMOC changes is still poorly documented.

Comment #8 - Despite the existence of some open issues, several publications (e.g., Lee et al., 2003; Anderson et al., 2009; Toggweiler et al., 2006; Tschumi et al., 2008; d'Orgeville et al., 2010; Lourantou et al., 2010; Sigman et al., 2010; Tschumi et al., 2011; Lee et al., 2011; Voigt et al., 2015) describe links between changes in AMOC and the Southern Hemisphere westerlies.

4) the opal flux in the Southern Ocean (Anderson et al. 2009) does not increase during HS2 and HS3.

Comment #9 - We are aware that Anderson et al.'s (2009) opal flux data (a proxy for the strength of Southern Ocean upwelling) did not change during the HS3 and HS2. However, this could be a temporal resolution issue. In core TN057-13-4PC, opal measurements show a mean temporal resolution of ca. 310 yr for HS1, but ca. 1550 yr for HS2 (this core does not reach HS3). In core E27-23, the mean temporal resolution of HS1 measurements is ca. 686 yr, but ca. 1550 yr for HS2, and ca. 1850 yr for HS3. HS1 opal measurements in core NBP9802-6PC have a mean temporal resolution of ca. 1715 yr, while HS2 measurements have a mean temporal resolution of ca. 3100 yr (this core does not reach HS3). Additionally, Anderson et al. (2009) suggest that such mechanism may be a common feature for other HS of the last glacial period.

As such the whole discussion section, as well as conclusion and abstract need to be rewritten.

Comment #10 - To account for the main topics raised by Referee #1 (but also Referee #2 and Dr. Schmittner) we have revised the necessary sections of our manuscript accordingly (please see above).

**1.2 Responses to Referee #2**

We thank Referee #2 for the constructive review of our manuscript.

Review of "Glacial d13C decreases in the western South Atlantic forced by millennial changes in Southern Ocean ventilation" by Campos et al.

The authors use the d13C of planktonic foraminifera to infer variability in the d13C of DIC in surface/thermocline waters in the SW Atlantic. One strength of the paper is the comprehensive review of the modern oceanographic setting. Another is that the planktonic time series are based on a sediment core with very high sedimentation rates, allowing for high resolution reconstruction of surface ocean d13C during Heinrich Stadial 2 and 3. It is also important that the authors used two different species to reconstruct d13C (one surface dwelling and another thermocline dwelling) to account for potential biases in habitat and vital effects that could overprint changes in the d13C of DIC. Given the high resolution and replicated nature of the record, the authors clearly show that this part of the Southwest Atlantic underwent significant changes in d13C during Heinrich Stadial 2 and 3.

The primary weakness of the paper is the interpretation of the d13C and sedimentation rate results. The authors are quick to assume that their d13C records reflect the input of light carbon from the Southern Ocean and neglect other possible explanations that are well documented in the published literature.

Comment #11 - We agree that other mechanisms could have also contributed to our records. Considering this comment and similar suggestions from Referee #1 and Dr. Schmittner, we have incorporated two additional hypotheses (see below) in the revised version of our manuscript, in order to have a more thorough and balanced discussion:

[revised manuscript text omitted]

In the conclusions we also inserted the two additional hypotheses described above and removed the mention of the Southern Ocean as the main source of the $CO_2$ outgassed to the atmosphere during HS3 and HS2.

Page 13, lines 14-18: we added *"(ii) Weakening of the global oceanic biological pump. A weak AMOC during HS3 and HS2 would promote an accumulation of $^{13}C$–depleted $CO_2$ in the upper water column of the South Atlantic. This accumulation would result in a negative anomaly of the $\delta^{13}C_{DIC}$ (as well as of the $\delta^{13}CO_{2atm}$) that in turn would be captured by the tests of planktonic foraminifera at our core site. We further suggest that changes in air–sea gas exchange could have contributed to the decreases in $\delta^{13}C$ via both mechanisms.".*

Page 13, line 20: we removed "*of the Southern Ocean*".

However, for the reasons stated below we still consider the possible role played by changes in Southern Ocean ventilation in producing our $\delta^{13}C$ anomalies:

iii. the mechanisms that invoke the Southern Ocean as one of the $CO_{2atm}$–sources during the last deglaciation are based on reconstructions as well as conceptual and numerical models that, to our understanding, have not yet been negated (Toggweiler et al., 2006; Anderson et al., 2009; Sigman et al., 2010; Tschumi et al., 2011; Lee et al., 2011; Burke and Robinson, 2012);

iv. model experiments that question the role of the Southern Ocean (e.g., Schmittner and Lund, 2015) were performed under preindustrial boundary conditions, which can significantly influence the results (e.g., Menviel et al., 2008), and do not constitute a firm negation of the Southern Ocean hypothesis.

The authors also invoke speculative connections between rainfall and sedimentation rate in the core as support for their climate interpretation, despite disagreement between the sedimentation rate and planktonic d13C patterns.

Comment #12 - We agree that the second HS2 peak in sedimentation rate and the respective planktonic $\delta^{13}C$ anomaly are not perfectly aligned in time. However, this apparent offset can be exclusively due to (i) the occurrence of $^{14}C$ plateaus during HS (e.g., Sarnthein et al., 2007; Franke et al., 2008), and/or (ii) the discretized way our age model was produced in relation to the "continuous" $\delta^{13}C$ measurements.

Finally, the authors neglect to mention much of the work at the Brazil Margin spanning the last deglaciation (including Heinrich Stadial 1) that is both relevant to their work and inconsistent with a Southern Ocean driver.

Comment #13 - We have included relevant publications from the Brazil Margin (e.g., Tessin and Lund, 2013; Lund et al. 2015) in the revised version of our manuscript together with a thorough and more balanced discussion (please see our Comment #11 above). However, the hypothesis that the Southern Ocean acts as a possible $CO_{2atm}$–source during HS has, to the best of our knowledge, not yet been proven incorrect and we still see the need to discuss it together with other possible explanations (please see our Comment #11 above).

Major issues:

While the prevailing view is that Southern Ocean outgassing drove surface ocean d13C anomalies during the last deglaciation, the authors also need to reference to Andreas Schmittner's work that shows that weakening of the AMOC can alter the preformed

nutrient budget of the global ocean and therefore the efficiency of the biological pump. Weakening of the biopump would preferentially leave light carbon in the surface and create negative d13C anomalies in multiple ocean basins (e.g. Schmittner, 2005; Schmittner and Galbraith, 2008).

Also, detailed reconstructions from the Brazil Margin show that benthic d13C and d18O changed late in the deglaciation, suggesting that the abyssal ocean was an unlikely source for the surface ocean anomalies during HS1 (Lund et al., 2015). Additional published records from the SE Atlantic shows a similar pattern, with a late deglacial response in the abyssal records (Waelbroeck et al., 2011; Roberts et al., 2016).

Comment #14 – Considering that the role of the weakening of the biological pump was also raised by Referee #1 and Dr. Schmittner, we have incorporated this hypothesis in the revised version of our manuscript (please see our Comment #11 above).

Our records complement the Brazil Margin depth transect (e.g., Tessin and Lund, 2013; Lund et al., 2015; Hertzberg et al., 2016) with upper water column high temporal resolution data that also agree with the model output of Schmittner and Lund (2015) by suggesting a decrease in $\delta^{13}C_{DIC}$ at the upper water column of the western South Atlantic.

In the revised version of our manuscript, we suggest that two processes may have contributed to the negative $\delta^{13}C$ anomalies in our records during HS3 and HS2: (i) strengthening of Southern Ocean deep water ventilation and (ii) weakening of the biological pump. Additionally, we suggest that air–sea gas exchange could have contributed to the observed $\delta^{13}C$ decreases.

Furthermore, intermediate depth d13C reconstructions from the Brazil Margin (Oppo and Horowitz, 2000; Curry and Oppo, 2005; Lund et al., 2015) imply that AAIW and SAMW were not carriers of a light carbon signal during HS1, which is inconsistent with the mechanism invoked by Campos et al. While the authors did a commendable job summarizing the literature pertaining the modern oceanographic setting, the introductory material includes surprisingly little information from previously published work along the Brazil Margin. As a result, the introductory material is incomplete and the lack of context limits the interpretation in the discussion section.

Comment #15 - To account to this additional mechanism in the revised manuscript we have inserted a new section describing the AMOC–induced weakening of the biological pump (please see our Comment #11 above).

The other main issue with the data interpretation is that the continental hydroclimate response isn't well supported by the sedimentation rate data. While sedimentation rates peak early in HS2, at about the same time as a planktonic d13C minimum, there is a second peak in sedimentation rate after HS2 that corresponds to a broad maximum in d13C. Furthermore, there is no peak in sedimentation rate during either HS3 or HS1. Taken as a whole, the sedimentation rate data therefore do not support a continental hydroclimate connection, which perhaps isn't surprising given the many factors that can influence sedimentation rates at a single core site.

Comment #16 - Please see our Comment #12 above. Additionally, although HS3 seems not to be related to a positive anomaly in sedimentation rate at our core site, HS1 and the Younger Dryas are both related to positive anomalies (of smaller amplitude during the Younger Dryas if compared to the anomalies during HS2 and HS1). The relatively high sea–level during HS3 and the Younger Dryas if compared to the intervening period (Waelbroeck et al., 2002) may have dampened a more significant anomaly in sedimentation rates during HS3 and the Younger Dryas (e.g., Lantzsch et al., 2014). This discussion has not been inserted in the revised manuscript because we have focused in HS3 and HS2.

Finally, the manuscript would benefit from inclusion of the planktonic d18O data. If there is a clear hydroclimate signal at this site, it could appear in the d18O data as intervals of unusually low d18O. The d18O data would also be informative for assessing the influence of air-sea gas exchange effects on the d13C signal. The d13C of surface ocean DIC is influenced by temperature dependent gas exchange, with a relationship of -0.1 per mil per deg C of warming (Broecker and Maier-Reimer, 1992; Lynch-Stieglitz et al., 1995). While such an effect is unlikely to explain the full magnitude of the d13C signals, it should be included in the discussion. Model results suggest that weakening of the AMOC warms the upper subtropical South Atlantic by 2-3 deg C (Marcotte et al., 2011) which would account for up to 0.3 per mil of the planktonic d13C anomalies.

Comment #17 - A similar topic was also raised by Referee #1. Please refer to our Comment #5 to Referee #1. Additionally, a section expanding the discussion on the possible role of air–sea gas exchange is included in the revised version of our manuscript. This mechanism is described in our Comment #11 above.

Regarding $\delta^{18}O$ data we inserted this data in Supplementary material (Figure 1). It is noteworthy that our foraminiferal $\delta^{18}O$ data do not present clear trends during HS3 and HS2. The $\delta^{18}O$ of planktonic foraminifera is markedly influenced by the temperature and seawater–$\delta^{18}O$ of the sea water where the tests calcified (e.g., Rohling and Cooke, 1999). The relationship between temperature and $\delta^{18}O$ of planktonic foraminifera is inversely proportional, and the relationship between seawater–$\delta^{18}O$ and $\delta^{18}O$ of planktonic foraminifera is directly proportional. Therefore, higher temperatures decrease foraminiferal $\delta^{18}O$, whereas higher seawater–$\delta^{18}O$ increases foraminiferal $\delta^{18}O$ (e.g., Rohling and Cooke, 1999).

For periods and regions of the global ocean where an increase (decrease) in temperature is associated to a decrease (increase) in seawater–$\delta^{18}O$, the $\delta^{18}O$ of planktonic foraminifera will clearly record such environmental changes (e.g., Guilderson and Pak, 2007). However, for those cases where temperature and seawater–$\delta^{18}O$ increase (decrease) simultaneously, these environmental changes may not be clearly recorded in foraminiferal $\delta^{18}O$. The last scenario seems to be the case for the upper water column under the influence of the Brazil Current for centennial-scale events of the Holocene (Chiessi et al., 2014), as well as for HS1 (Chiessi et al., 2015), and possibly also for HS3 and HS2 (given the lack of regional SST reconstructions for HS3 and HS2).

Pages 11-12, lines 29-2: "*The $\delta^{18}O$ records from G. ruber w and G. inflata from our core (Supplementary material Figure 1) should partially reflect changes in water temperature (ca. -0.22‰ per 1 °C; e.g., Mulitza et al, 2003), but show no clear trends across HS3 and HS2. While temperature changes might be partially obscured in the foraminiferal $\delta^{18}O$ records by the influence of synchronous changes in seawater-$\delta^{18}O$, as has been hypothesized for the Holocene (Chiessi et al., 2014) and HS1 (Chiessi et al., 2015) in the western Atlantic, we consider it unlikely that temperature changes of the above magnitude would be completely masked.*".

Minor issues:

Page 6, Line 12: Here the authors should consider the possibility that weakening of the biological pump could have produced d13C anomalies not only locally, but on a larger spatial scales.

Comment #18 - Agreed. We have changed the revised version of our manuscript accordingly, but have not inserted the suggested change at this passage. Rather, we inserted the suggested change to pages 10-11, lines 10-11 (please see our Comment #11 above).

Page 6, Line 20: It is important to mention that published mode and intermediate water records from the Brazil Margin either show a positive d13C anomaly during HS1 or a delayed negative d13C response that is inconsistent the light carbon being transported northward via mode and intermediate water.

Comment #19 - Agreed (please see our Comment #11 above). We have revised the manuscript accordingly. Additionally, we also incorporate two more hypotheses (i.e., 5.2 Millennial–scale changes: AMOC–induced weakening of the biological pump, and 5.3 Millennial–scale changes: the role of air–sea gas exchange) to the revised version of our manuscript, in order to have a more thorough and balanced discussion.

Page 7, Line 5: The authors seem to take it as a given that Southern Hemisphere westerlies drives greater upwelling of deep waters when it actuality there is limited data to support such a link. Please qualify these types of statements to reflect the limited constraints that exist.

Comment #20 – Agreed (please see our Comment #8 to Referee #1). We have revised our manuscript accordingly.

Page 7, Line 9: See comments above that published intermediate depth records from the Brazil Margin don't show clear negative carbon isotope anomalies during HS1, which is relevant to whether a similar process occurred during HS2 and HS3.

Comment #21 - Agreed (please see our Comment #11 above). We have revised our manuscript accordingly. Additionally we also incorporate two more hypotheses (i.e., 5.2 Millennial–scale changes: AMOC–induced weakening of the biological pump, and 5.3 Millennial–scale changes: the role of air–sea gas exchange) to the revised version of our manuscript, in order to have a more thorough and balanced discussion.

Figure 4: If the authors are going to pursue the sedimentation rate-hydoclimate link, then the sedimentation rates should also be included in this figure for comparison to the d13C. The d18O records should also be included to assess the potential impact of sea surface temperature on d13C.

Comment #22 - Agreed. We have inserted the sedimentation rate record in Figure 5 of the revised version of our manuscript. Regarding the $\delta^{18}O$ data, please refer to our Comment #5 to Referee #1 and our Comment #17 above.

Figure 5: There are several time series in this plot that aren't essential for the discussion, such as the Iberian Margin SST record, the EDML d18O time series, and the Siple Dome CO2.

Comment #23 - We agree that removing the $\delta^{18}O$ EDML curve (EPICA Community Members, 2006) would not harm the main focus of our manuscript. In addition, we have also removed the Taylor Dome $\delta^{13}CO_{2atm}$ curve (Smith et al., 1999) for the reason mentioned in our Comment #7 to Referee #1.

However, we would prefer not to delete the following curves: (i) the Iberian Margin SST curve (Bard, 2002) because it shows a "w–structure" during HS2 that gives support to the proposed "w–structure" of HS2; and (ii) the Siple Dome $CO_{2atm}$ (Ahn and Brook, 2014) due to its fundamental importance to our manuscript (please see our Comment #6 to Referee #1).

**1.3 Response to Dr. Schmittner**

We thank Dr. Schmittner for his constructive comment on our manuscript. To facilitate the discussion, we have copied his comment below in black and inserted our response in green.

I suggest to the authors to consider our relevant recent modeling work, which suggests a different mechanism for the CO2 increase during H-events. As shown in Schmittner and Galbraith (2008, Nature, 456, 373-376, doi:10.1038/nature07531) an AMOC shutdown causes a decrease of the efficiency of the biological pump, which leads to an increase in atmospheric CO2 consistent in both amplitude and rate-of-change with ice core observations. Schmittner and Lund (2015; Climate of the Past, 11, 135-152,

doi:10.5194/cp-11-135-2015) show that this leads to a decrease of surface ocean (and atmospheric) d13C that is particularly strong (more than 0.5 permil) in the South Atlantic (their Fig. 5G).

Comment #1 - A similar observation was also raised by Referee #1 and Referee #2. Please refer to our Comment #1 to Referee #1 or Comment #11 to Referee #2.

**2. Additional relevant changes performed in the manuscript**

After implementing the changes suggested by the Referees and by Dr. Schmittner, we identified the need to refine specific passages of the manuscript. Those relevant changes are shown below in italic green in quotes.

**2.1 Changes performed in the title**

Page 1, lines 1-2: the title was changed to "*$\delta^{13}C$ decreases in the upper western South Atlantic during Heinrich Stadials 3 and 2*".

**2.2 Relevant changes performed in the abstract**

Page 1, lines 12-13: we added "*($CO_{2atm}$) and decreases of its stable carbon isotopic ratios ($\delta^{13}C$), i.e., $\delta^{13}CO_{2atm}$*".

Page 1, line 13: we exchanged the term "*Southern Ocean*" by "*ocean*".

Page 1, lines 16-17: we removed "*intensification in Southern Ocean deep water ventilation presumably associated with a*".

Page 1, lines 18-21: we added "*and the consequent increase (decrease) in $CO_{2atm}$ ($\delta^{13}CO_{2atm}$). We hypothesise two mechanisms that could account for the decreases observed in our records, namely strengthening of Southern Ocean deep water ventilation and weakening of the biological pump. Additionally, we suggest that air–sea gas exchange could have contributed to the observed $\delta^{13}C$ decreases.*".

Page 1, lines 21-22: we removed "*After reaching the upper water column of the Southern Ocean, the $\delta^{13}C$ depletion would be transferred equatorward via central and thermocline waters.*".

Page 1, line 24: we removed "*from the Southern Ocean*".

Page 1, line 29-30: we exchanged the term "*Southern Ocean*" by "*Atlantic Meridional Overturning Circulation*".

**2.3 Relevant changes performed in the introduction**

Page 2, lines 10-11: we added "*This increase in $CO_{2atm}$ was accompanied by a decrease of its stable carbon isotopic composition ($\delta^{13}CO_{2atm}$) (Eggleston et al., 2016).*".

Page 2, lines 11-12: we added "*and the associated $\delta^{13}CO_{2atm}$ decrease*"

Page 2, lines 20-26: we added "*However, model experiments (e.g., Schmittner and Galbraith, 2008; Schmittner and Lund, 2015) and records from different ocean basins (e.g., Tessin and Lund, 2013; Lund et al., 2015; Curry and Oppo, 2005; Hertzberg et al., 2016) suggest that the increase in $CO_{2atm}$ and decrease in $\delta^{13}CO_{2atm}$ during HS1 are instead related to the weakening of the global oceanic biological pump and the consequent accumulation of $^{13}C$–depleted $CO_2$ in the upper water column. Such anomalously low–$\delta^{13}CO_2$ would then be outgassed to the atmosphere. Since isotopic fractionation between reservoirs is temperature–dependent the air–sea gas exchange during HS1 could have additionally modified $\delta^{13}CO_{2atm}$ (Lynch-Stieglitz et al., 1995).*".

Pages 2-3, lines 27-1: we removed "*The CDW forms from mixing of North Atlantic Deep Water (NADW), Indian Deep Water (IDW) and Pacific Deep Water (PDW) and upwells to the south of the Antarctic Polar Front driven by the prevailing westerly winds (Marshall and Speer, 2012). Therefore, the $\delta^{13}C$ signal of CDW (ca. 0.4‰) (Kroopnick, 1985) lies between that of NADW (ca. 1‰) (Kroopnick, 1985) and IDW/PDW (ca. 0.2 to -0.2‰) (Kroopnick, 1985) (Oppo and Fairbanks, 1987; Charles and Fairbanks, 1992). During periods of weak AMOC the influence of NADW on the Southern Ocean is reduced (Charles and Fairbanks, 1992), and the $\delta^{13}C$ of CDW should decrease since this water mass would have a relatively larger contribution from*

*low–$\delta^{13}$C IDW and PDW. Thus, the strengthening in the Southern Ocean upwelling would bring old, respired $^{13}$C–depleted waters to the surface that would then be transferred equatorward via central and thermocline waters (Spero and Lea, 2002).".*

Page 3, lines 1-2: we added "*The reduction of the upper water column $\delta^{13}$C caused by one or both of the*".

Page 3, lines 2-3: we removed "*(i.e., a weak AMOC leading to a strengthening of the Southern Ocean upwelling, a $\delta^{13}$C depletion and an increased outgassing of $CO_2$ to the atmosphere)*".

Page 3, lines 11-12: we removed "*most likely caused by millennial–scale strengthening of Southern Ocean upwelling*".

**2.4 Relevant changes performed in the regional settings**

Page 3, line 24: we added "*In the study area*".

Page 3, lines 24-25: we added "*and Antarctic Intermediate Water (AAIW)*".

Page 4, lines 12-13: we added "*the dissolved inorganic carbon $\delta^{13}$C ($\delta^{13}C_{DIC}$) in*".

Page 4, line 16: we added "*In the southwest South Atlantic*".

**2.5 Relevant changes performed in the materials and methods**

Page 5, lines 19-22: we removed "*Default parameter settings were used, except for mem.mean (set to 0.4) and acc.shape (set to 0.5). Ages are modelled as drawn from a t–distribution, with 9 degrees of freedom (t.a=9, t.b=10). 1,000 age–depth realizations were used to estimate mean age and 95 % confidence intervals at 0.5 cm resolution (Fig. 3).*".

Page 5, lines 22-25: we added "*With the exception of mem.mean (set to 0.4) and acc.shape (set to 0.5), default parameters were used. Radiocarbon ages were assumed to be t-distributed with 9 degrees of freedom (t.a.=9, t.b=10). Mean ages and 95% error*

*margins were estimated from 10,000 downcore age-depth realizations at 0.5 cm resolution (Fig. 3).".*

Page 5, lines 24: in order to improve our age model we increased the number of age-depth realizations modeled within the software Bacon 2.2 from "*1,000*" to "*10,000*".

**2.6 Relevant changes performed in the results**

Page 6, line 7: the section title was changed to "*Age model and sedimentation rates*".

Page 6, line 8: we exchanged "*32.6*" by "*32*".

Page 6, line 8: we exchanged "*5.7*" by "*6*".

Page 6, line 9: we exchanged "*3.8*" by "*5*".

Page 6, line 9: we exchanged "*111*" by "*91*".

Page 6, lines 11-12: we removed "*and received special attention due to the higher sedimentation rate which provides increased temporal resolution*".

Page 6, line 14: the section title was changed to "*Stable carbon isotope values of G. ruber and G. inflata*".

Page 6, line 15: we exchanged "*32.6*" by "*32*".

Page 6, line 15: we exchanged "*28.5*" by "*28.2*".

Page 6, line 16: we exchanged "*24.8*" by "*24.9*".

Page 6, line 17: we exchanged "*27*" by "*27.2*".

Page 6, line 18: we exchanged "*30.6*" by "*29.3*".

Page 6, line 18: we exchanged "*30.4*" by "*29.1*".

Page 6, line 23: we exchanged "*32.5*" by "*31.5*".

Page 6, line 23: we exchanged "*30.6*" by "*29.3*".

Page 6, line 23: we exchanged "*29.8*" by "*28.8*".

Page 6, line 23: we exchanged "*28.3*" by "*28*".

**2.7 Relevant changes performed in the discussion**

Page 7, lines 4-12: we added "*Concomitantly, a weak AMOC was described based on $^{231}Pa/^{230}Th$ records from the Bermuda Rise (ODP Site 1063, 33.7° N, 57.6° W) (Lippold et al., 2009) (Fig. 5d). Both events are also marked by pulses of ice-rafted debris (IRD) (MD99-2331, 42.2° N, 9.7° W) (Eynaud et al., 2009) and by decreases in SST (SU8118 and MD952042, 37.5° N, 10.1° W) (Bard, 2000) in the north–eastern North Atlantic (Iberian Margin). The Greenland GISP2 ice core (72.6° N, 38.5° W) shows synchronous increases in $Ca^{+2}$, indicating changes in atmospheric circulation over Greenland (Mayewski et al., 1997) (Fig. 5a, b, c). It is noteworthy that the four records (i.e., Fig. 5a, b, c, d) mentioned above also show a "w–structure" during HS2, similar to the one shown in our $\delta^{13}C$ records. The IRD (Eynaud et al., 2009) and $Ca^{+2}$ (Mayewski et al., 1997) records also show a "w–structure" similar to ours during HS3.*"

Page 7, line 13: we added "*local*".

Page 7, lines 26-29: we added "*We propose two primary mechanisms to explain our $\delta^{13}C$ decreases: (i) changes in the strength of Southern Ocean deep water ventilation (detailed in section 5.1), and (ii) the weakening of the global oceanic biological pump (detailed in section 5.2). Additionally, air–sea gas exchange may have acted as a secondary factor contributing to our $\delta^{13}C$ decreases (detailed in section 5.3).*".

Pages 7-8, lines 30-6: we removed "*A pervasive feature of planktonic foraminiferal $\delta^{13}C$ records in the Indo–Pacific Ocean (Spero and Lea, 2002), Southern Ocean (Ninnemann and Charles, 1997), and South Atlantic Ocean (Oppo and Fairbanks, 1989) is a negative excursion during HS1. Ninnemann and Charles (1997) suggested that the source for this signal is in the Southern Ocean. They further proposed that the anomaly is related to the transfer of a preformed $\delta^{13}C$ signal from the Southern Ocean via SAMW and/or AAIW. A low–density type of SAMW actually contributes to SACW that spreads into the South Atlantic (Stramma and England, 1999). Additionally, AAIW also influences SACW through vigorous eddy mixing at the Brazil/Malvinas Confluence*

*(Piola and Georgi, 1982). Thus, SACW represents a potential conduit for the $\delta^{13}C$ signal from the sub–Antarctic region to the subtropical South Atlantic (Fig. 1 and 2). Therefore, we propose that the negative excursions in our $\delta^{13}C$ records are related to the transfer of a preformed $\delta^{13}C$ signal from the subantarctic zone to the western South Atlantic via central and thermocline waters.".*

Page 8, lines 7-8: the section title was changed to "*Millennial–scale changes: AMOC–induced strengthening of Southern Ocean deep water ventilation*".

Page 8, lines 9-20: we added "*A negative excursion during HS1 was described in planktonic foraminiferal $\delta^{13}C$ records from the Indo–Pacific Ocean (Spero and Lea, 2002), Southern Ocean (Ninnemann and Charles, 1997), and South Atlantic Ocean (Oppo and Fairbanks, 1989). Ninnemann and Charles (1997) suggested that the source for this signal was the Southern Ocean.*

*In the Southern Ocean CDW forms from mixing of NADW, Indian Deep Water (IDW) and Pacific Deep Water (PDW) and upwells to the south of the Antarctic Polar Front driven by the prevailing westerly winds (Marshall and Speer, 2012). Therefore, the $\delta^{13}C$ signature of CDW (ca. 0.4‰) (Kroopnick, 1985) lies between that of NADW (ca. 1‰) (Kroopnick, 1985) and IDW/PDW (ca. 0.2 to -0.2‰) (Kroopnick, 1985) (Oppo and Fairbanks, 1987; Charles and Fairbanks, 1992). During periods of weak AMOC the inflow of NADW to the Southern Ocean is reduced (Charles and Fairbanks, 1992), and the $\delta^{13}C$ of CDW should decrease since the latter would have a relatively larger contribution from low–$\delta^{13}C$ IDW and PDW (Spero and Lea, 2002).".*

Page 8, lines 23-31: we removed "*Concomitantly to the periods of low $\delta^{13}C$ in our records a weak AMOC was described based on $^{231}Pa/^{230}Th$ records from the Bermuda Rise (ODP Site 1063, 33.7° N, 57.6° W) (Lippold et al., 2009) that characterize HS3 and HS2 (Fig. 5d). Both events are also marked by pulses of ice-rafted debris (IRD) (MD99-2331, 42.2° N, 9.7° W) (Eynaud et al., 2009) and by decreases in SST (SU8118 and MD952042, 37.5° N, 10.1° W) (Bard, 2002) in the north–eastern North Atlantic (Iberian Margin). The Greenland GISP2 ice core (72.6° N, 38.5° W) shows synchronous increases in $Ca^{+2}$, indicating changes in atmospheric circulation over Greenland (Mayewski et al., 1997) (Fig. 5a, b, c). It is worth to note that the four records (i.e., Fig. 5a, b, c, d) mentioned above also show a w–structure during HS2,*

*similar to the one shown in our $\delta^{13}C$ records. The IRD (Eynaud et al., 2009) and $Ca^{+2}$ (Mayewski et al., 1997) records also show a w–structure similar to ours during HS3.".*

Page 9, lines 5-7: we added "*A recent model experiment (Bauska et al., 2016) corroborates this hypothesis by showing that stronger Southern Ocean upwelling would promote a weakening of the biological pump in the Southern Ocean.*".

Page 9, lines 11-13: we added "*Actually, a low–density type of SAMW contributes to SACW that spreads into the South Atlantic (Stramma and England, 1999). Additionally, AAIW also influences SACW through vigorous eddy mixing at the Brazil/Malvinas Confluence (Piola and Georgi, 1982).*".

Page 9, lines 15-17: we removed "*However, we cannot exclude the possibility that the upwelled low–$\delta^{13}C$ respired $CO_2$ could have been first outgassed from the Southern Ocean, and then re–dissolved into the ocean via air–sea exchanges at the formation regions of SACW and TW, eventually reaching the upper water column at our core site.*".

Page 9, lines 23-26: we removed "*Millennial–scale changes of the Southern Ocean temperature and deep water ventilation also led to the increase in $CO_{2atm}$ (Spero and Lea, 2002; Ahn and Brook, 2008; Ahn and Brook, 2014; Gottschalk et al., 2015). During HS3 and HS2, positive peaks in $CO_{2atm}$ (Siple Dome, 81.7° S, 148.8° W) (Ahn and Brook, 2014) (Fig. 5k) were described to be synchronous to circum–Antarctic warming (EPICA Community Members, 2006) (Fig. 5i), and most likely have originated from*".

Page 9, lines 29-30: we added "*and the consequent increase in $CO_{2atm}$ (Siple Dome, 81.7° S, 148.8° W) (Ahn and Brook, 2014) (Fig. 5j)*".

Page 9, lines 31-32: we removed "*even the $\delta^{13}C$ record of $CO_{2atm}$ at Taylor Dome (77.8° S, 158.7° E) on Antarctica shows a decrease during HS2 (Smith et al., 1999) (Fig. 5j).*".

Page 10, lines 2-3: we removed "*An alternative explanation relates to a possible time lag between the weakening of the AMOC and the increase in $CO_{2atm}$ (Ahn and Brook, 2014)*".

Page 10, lines 4-5: we added *"Therefore, the negative excursions in our $\delta^{13}C$ records could be related to the transfer of a preformed $\delta^{13}C$ signal from the subantarctic zone to the western South Atlantic via central and thermocline waters."*.

Page 10, lines 6-9: we removed *"Thus, our records are consistent with the hypothesis that the increase in $CO_{2atm}$ during abrupt millennial–scale climate change events of the last glacial period is originated by ocean processes (Smith et al., 1999; Ahn and Brook, 2008; Bereiter et al., 2012; Ahn and Brook, 2014) and is most likely related to a weak AMOC and associated strengthened Southern Ocean upwelling."*.

Page 12, line 3: the section title was changed to *"Changes in continental climate"*.

Page 13, lines 3-5: we removed *"However, we discard this possibility because the expected signal of stronger local primary productivity on planktonic foraminiferal $\delta^{13}C$ would be opposite to the one observed at GeoB6212-1 (Mulitza et al., 1999)."*.

**2.9 Relevant changes performed in the acknowledgements**

Page 13, lines 28-1: we added *"We thank two anonymous reviewers and A. Schmittner for constructive comments that greatly improved this manuscript."*.

**2.10 Relevant changes performed in table 1**

Page 22, table line 1: we exchanged *"5694"* by *"5929"*.

Page 22, table line 2: we exchanged *"10112"* by *"10047"*.

Page 22, table line 3: we exchanged *"11345"* by *"11395"*.

Page 22, table line 4: we exchanged *"14374"* by *"14342"*.

Page 22, table line 5: we exchanged *"15179"* by *"15247"*.

Page 22, table line 6: we exchanged *"18616"* by *"18620"*.

Page 22, table line 7: we exchanged *"20731"* by *"20748"*.

Page 22, table line 8: we exchanged *"21843"* by *"21834"*.

**2.11 Relevant changes performed in the figures**

[Figure]

[Figure]

Page 29: we exchanged Figure 5 by

[Figure]

Page 30, lines 16-17: we added "*sedimentation rates from marine sediment core GeoB6212-1 collected in the western South Atlantic at 32.4° S, 50.1° W (this study)*".

Page 30, lines 17-20: we removed "*atmospheric temperature over Antarctica indicated by EPICA Dronning Maud Land (EPICA Community Members, 2006) $\delta^{18}O$ (higher values indicate warmer conditions) plotted versus its original chronology at 75° S, 0° E; (j) $\delta^{13}C$ of atmospheric $CO_2$ ($\delta^{13}CO_{2atm}$) from Taylor Dome (lower values indicate increased input to the atmosphere of $\delta^{13}C$–depleted respired $CO_2$) at 77.8° S, 158.7° E (Smith et al., 1999); (k)*".

In this section we inserted the marked up revised version of the manuscript. The changes performed to the manuscript are marked up in purple and green.

[revised manuscript text omitted]

---

## Author Response (AR2)

**1. Point-by-point response and relevant changes performed in the manuscript**

In this section we inserted a point-by-point response to the Referees together with all relevant changes performed in the manuscript. To facilitate the review, we copied the Referees´ comments below in black and inserted our comments in green. The modifications performed to the manuscript are shown in italic green in quotes.

**1.1 Responses to Referee #1**

Campos et al., significantly improved the manuscript from the first draft and I appreciate the effort put into the answer to previous comments. Please find below a few comments that should be taken into account before publication.

We thank Referee #1 for the positive review of our manuscript.

1) The new figure 5 is very nice, and the similarity between d13C GeoB6212-1 and atm. pCO2 is striking. This also raises the issue of the interpretation given by the authors: D13C decreases in phase with pCO2 decrease, contrarily to the hypothesis given.

Please note that some 3-dimensional numerical experiments of an AMOC shut-down have been performed with prognostic oceanic and atmospheric d13C. For example, in Menviel et al., 2015, you can see that both the Bern3D and LOVECLIM suggest that NADW shutdown leads to a surface d13C decrease on the Brazilian margin even without enhanced Southern Ocean upwelling. An AMOC shutdown without enhanced Southern Ocean upwelling leads to a small pCO2 decrease (Menviel et al., 2014).

Comment #1 - There is indeed a delay between the minima in our $\delta^{13}$C records and the maxima in the Siple Dome $CO_{2atm}$ record (Ahn and Brook, 2014). We clearly recognize these offsets in our manuscript (page 8, lines 11-14) and attribute them to uncertainties in the respective age models. It is noteworthy that Ahn and Brook (2014) show the best resolved $CO_{2atm}$ record for HS3 and HS2, and that record shows an increase in $CO_{2atm}$ during HS3 and HS2 (as is highlighted by red arrows in the original publication - please see Fig. 1c from Ahn and Brook (2014)).

2) Abstract and P11, L20-22: With the currently proposed hypothesis (AMOC shutdown leads to enhanced SO upwelling, which decreases SO d13C, releases low d13C into the atmosphere and induce surface d13C decrease on the Brazilian margin), I don't see how the d13C records presented here add evidence to the fact that atmospheric CO2 increase during HS2 and HS3 could have originated from the ocean. Terrestrial carbon release would also increase pCO2, decrease d13CO2 and thus oceanic d13C.

Comment #2 - We agree that based on our data alone we cannot exclude the role of terrestrial carbon release as a potential source for the increase in $CO_{2atm}$. However, the two primary mechanisms (i.e., (i) AMOC–induced strengthening of Southern Ocean deep water ventilation, and (ii) AMOC–induced weakening of the biological pump) invoked in our manuscript to explain the $\delta^{13}$C decreases in our records are both related to the oceanic release of $CO_2$ to the atmosphere. The full rationale for both mechanisms is

described in sections 5.1 and 5.2 of our manuscript, and is supported by previously published records and model experiments.  To prevent from misleading the reader, we changed the sentence in the Abstract, as follows (a similar cautionary formulation was also used in the other sentence mentioned by Referee #1):

Page 1, lines 19-20: we added "*Together with other lines of evidence*".

3) In several part of the paper, the authors suggest that similar mechanisms/teleconnctions were at play during all Heinrich stadials (HS2, HS3 often compared to HS1). I understand the rationale, but this is not necessarily true, particularly in the case of HS1, which occurred during the deglaciation … For example, they assume that d13CO2 decreased during HS2 and HS3. It is quite possible, but not yet firmly shown. The very nice d13CO2 record from E'ggleston et al., 2016, unfortunately cannot give that information. They note on p14: "Note that while Heinrich event 2 may also have left a small imprint in our $\delta$13C(atm) record, an unambiguous signal of the weaker Heinrich event 3 cannot be discerned. However, the low resolution in the $\delta$13C(atm) record at that time precludes a final conclusion."

Comment#3 - We agree that changes in $\delta^{13}CO_{2atm}$ for HS3 are still inconclusive because of the lack of the necessary temporal resolution in the available records, despite the evidence from model experiments (Schmittner and Galbraith, 2008; Schmittner and Lund, 2015). Thus, we no longer mention that HS3 was accompanied by a decrease in $\delta^{13}CO_{2atm}$.

Page 2, line 6: we added "*at least for some HS*".

Page 9, line 28: we removed "*HS3 and*".

4) P7, L.26-28: Bauska et al., 2016 present a high-resolution high-quality atmospheric d13CO2 of the last deglaciation, but the modeling experiments performed are extremely idealized and performed with a simple box model. Please note that the impact of changes in southern hemispheric westerlies and/or changes in Antarctic Bottom Water formation on oceanic and atmospheric d13C has been explored using 3-dimensional ocean models including OGCM (e.g. Tschumi et al., 2011; Menviel et al., 2015).

Comment#4 - Agreed. We inserted the references mentioned by Referee #1.

Page 7, line 27: we added "*Model experiments (Tschumi et al., 2011; Menviel et al., 2015*".

**1.2 Responses to Referee #2**

Overall the authors have addressed most of the major issues raised by the reviewers and the manuscript is close to ready for publication. However, there remain several points that still require attention:

We thank Referee #2 for the positive review of our manuscript.

Page 6, Line 12: Air sea gas exchange very likely played a role in creating the surface ocean d13C anomalies. For example, a large fraction of the modeled decrease in d13C in subtropical gyre locations is due to exchange with a d13C depleted atmosphere (see Figure 6A in Schmittner and Lund, 2015).

Comment #1 - We agree that air-sea gas exchange may have played a role on setting the $\delta^{13}$C anomalies in our records and section 5.3 of our manuscript is entirely devoted to this topic. (Please note that we found no allusion for this topic on the page/lines mentioned by Referee #2).

Page 2, Line 10: While the surface ocean d13C anomalies require a global-scale driver, perhaps via the Southern Ocean, the high resolution intermediate depth records compiled by Hertzberg et al. (2016) show that d13C increased during HS1, which is inconsistent with transport of light carbon by AAIW.

Comment #2 - Indeed, intermediate waters in the western South Atlantic showed an increase in $\delta^{13}$C during HS1, and we clearly mention it in our manuscript (page 9, lines 3-6), citing Tessin and Lund (2013), Lund et al. (2015), Curry and Oppo (2005), and Hertzberg et al. (2016). Moreover, in section 5.1 (more specifically on page 7, lines 29-32, and page 8, lines 1-4) of our manuscript, we suggest that the low $\delta^{13}$C signal could have been transferred to SACW at its region of formation, via SAMW and, to some extent, via AAIW (but, in this case, only up to the Brazil/Malvinas Confluence). (Please note that we found no allusion for this topic on the page/lines mentioned by Referee #2).

Page 2, Line 20: Note that temperature change is not required... an isotopically light surface ocean will cause the atmosphere to be lighter (assuming constant gas exchange rates), with the temperature or wind-driven gas exchange effects acting to modify the isotopic offset between the surface ocean and atmosphere.

Comment #3 - Agreed. An accumulation of $^{13}$C–depleted $CO_2$ in the upper water column will produce a reduction in $\delta^{13}CO_{2atm}$ via air–sea gas exchange. The temperature effect will act as an additional modifier promoting changes in $\delta^{13}CO_{2atm}$.

Page 2, line 19: we removed "*be outgassed to*".

Page 2, line 19: we added "*via air–sea gas exchange*".

Page 2, lines 19-21: we removed "*Since isotopic fractionation between reservoirs is temperature–dependent the air-sea gas exchange during HS1 could have additionally modified* $\delta^{13}CO_{2atm}$ *(Lynch-Stieglitz et al., 1995).*".

Page 4, Line 6: A minor detail, but air-sea exchange influences d13C of DIC, which then may be incorporated into the foraminiferal shell, with modification by vital effects.

Comment #4 - Agreed. All factors mentioned in page 4, line 6, i.e., calcification temperature, carbonate ion concentration, symbiont activity and air–sea gas exchange can

be incorporated into foraminiferal shell with modification by vital effects. We have changed this passage accordingly.

Page 4, line 7: we added "*with modification by vital effects*".

Page 4, Line 27: What is the interpretation of the sand lenses? Are they turbidites? If so, is there any evidence of anomalous stable isotope results or sedimentation rates above and below the sand layers?

Comment #5 - The sand lenses could indeed be turbidites and to avoid bias in the interpretation of our results we did not sampled these intervals. Additionally, the $\delta^{13}$C and sedimentation rate were not anomalous near these intervals.

Page 5, Line 3-7: Did the authors use mixed planktonic or individual species for the radiocarbon dates? If mixed, were they surface mixed layer or thermocline bugs, or both? What is the basis for assuming the 400 year surface water reservoir age? Is this because surface mixed layer bugs were used for 14C analysis? What is the assumed reservoir age error for the age calibration?

Comment #6 - Table 1 of our manuscript shows in which depths we used individual species, i.e., *G. ruber* (3 radiocarbon ages) and which ones required the use of mixed planktonic foraminifera (11 radiocarbon ages). For these 11 radiocarbon ages we were forced to pool together both mixed layer and thermocline species.

Page 5, lines 2-3: we added: "(*mixed layer and thermocline species)*".

Since we have used a terrestrial calibration curve within the software Bacon 2.2, i.e. IntCal13 (Reimer et al., 2013), it was necessary to apply a correction for reservoir age. This is expressed in the statement (page 5, lines 5-7) "All radiocarbon ages were calibrated with the calibration curve IntCal13 (Reimer et al., 2013) with the software Bacon 2.2 (Blaauw and Christen, 2011). A marine reservoir correction of 400 years was applied with associated error of 100 years (Bard, 1988).".

We chose not to apply a larger reservoir correction than the one suggested for the mixed layer between 40°N and 40°S by Bard (1988) for the samples also containing thermocline species of planktonic foraminifera because the proportion of mixed layer and thermocline species was not determined. However, the 100 yr uncertainty of the reservoir age definitely accommodates any plausible reservoir correction (Franke et al., 2008).

Regarding the reservoir age error, we used an error of 100 yr and added this information to the manuscript.

Page 5, line 7: we added "*with associated error of 100 years*".

Page 6, Line 16: There are two clear negative d13C anomalies during HS2 but the pattern is murkier prior to HS2. While there is a small anomaly at the end of HS3, there is a much larger negative excursion between HS3 and HS2 that doesn't fit the assumed pattern of low d13C during Heinrich stadial events. The authors need to acknowledge this

inconsistency and briefly discuss what may explain the large negative anomaly between HS3 and HS2.

Comment #7 - We agree that the "w-structure" in the $\delta^{13}C$ records fit well into HS2, but only partially within HS3. We assume that this is due to age model uncertainties and briefly discussed it in the manuscript.

Page 6, lines 16-17: we added: "*event HS2 and, although the slight offset that is attributed to age model uncertainties, also with*".

Page 7, Line 20: While it is true that the d13C of CDW should decrease with weakening of NADW, there are several additional factors that will influence the d13C tracer field of the ocean interior, related to changes in both preformed and remineralized d13C. For example, the model results of Schmittner and Lund (2015) suggest that remineralization plays the primary role in setting d13C in below 500m water depth in the South Atlantic, Indian, and Pacific Oceans following shutdown of the AMOC (see Figures 5 and 6, Schmittner and Lund, 2015).

Comment #8 - Agreed. In the manuscript, we address these two mechanisms that could account for the decreases observed in our $\delta^{13}C$ records, i.e., the role of the Southern Ocean deep water ventilation (section 5.1), and the role of the biological pump (section 5.2). Section 5.2 specifically deals with the topic raised by Referee #2, i.e., preformed and remineralized $\delta^{13}C$ in the interior ocean.

Page 7, Line 24: Please qualify that the large Southern Ocean upwelling response occurs only in models with low resolution. Higher resolution models show only minor upwelling sensitivity to large changes in prescribed wind stress in the Southern Ocean, mainly due to eddy compensation (e.g. Farneti and Delworth, Journal of Physical Oceanography, v. 40, 2010).

Comment #9 - Agreed.

Page 7, lines 23-24: we added "*models of low resolution suggest that*".

Page 8, Line 7: This is somewhat selective presentation of evidence... as mentioned above, four separate intermediate depth sites show higher d13C during HS1, which is inconsistent with transport of isotopically light carbon via mode and intermediate waters (Hertzberg et al., 2016).

Comment #10 - Agreed. To prevent misleading the reader, we deleted this sentence.

Page 8, lines 4-7: we removed "*Hendry et al. (2012) describe higher seawater concentrations of Si(OH)4 through silicon isotope composition of sponge spicules (benthic organisms) at intermediate water depths (1048 m water depths) of the western South Atlantic (ca. 27° S) close to our core site during abrupt millennial–scale climate change events, suggesting that the preformed signal from the Southern Ocean could indeed reach subtropical latitudes in the South Atlantic.*".

Page 9, Line 20, Line 23: The d13C of DIC in the surface ocean IS affected by air-sea gas exchange and this would have also been the case during HS events, the question is to what extent. While it is true that the surface ocean rarely reaches equilibrium with the atmosphere, this does not mean the surface ocean doesn't respond to atmospheric variation, especially places like subtropical gyres where there is greater time for equilibration with the atmosphere.

Comment #11 - Agreed.

Page 9, lines 20-21: we added: "*(especially in subtropical gyres, because the longer water residence time in these regions)*".

Page 10, Line 3: Please include best estimate of how much temperature may have affected surface ocean d13C (assuming full equilibration with atmosphere, so it would be a maximum effect).

Comment #12 - The statements mentioned by Referee #2, i.e., "If temperature was the dominant driver, unrealistic changes between 5 and 13 °C would be required to explain the full amplitudes of the $\delta^{13}$C variations.", is not related to the estimates of how much the temperature may have affected surface ocean $\delta^{13}$C, but what should be the range of temperature increase to explain the range of decrease observed in our $\delta^{13}$C records. If our $\delta^{13}$C decreases ranged around 0.5-1.3 ‰, thus the range of increase in temperature required should be around 5-13 ºC, considering a reduction of 0.1 ‰ per °C (Broecker and Maier-Reimer, 1992). However, no high temporal resolution sea surface temperature record is available for HS3 and HS2 in the area of interest of the South Atlantic. Still, nearby records (Barker et al., 2009; Chiessi et al., 2015) show values not larger than 3 ºC for HS1, that showed a notably stronger sea surface temperature anomaly if compared to HS3 and HS2.

Page 10, Line 9: The relationship between sedimentation rate and d13C is inconsistent. While low d13C at the beginning of HS2 corresponds to high sedimentation rate, the even more negative d13C anomaly in mid-HS2 occurs during a relative peak in sedimentation. Also, there is no clear changes in sed. rate during HS3, which should be the case is southward ITCZ migration and greater riverine input of terrigenous material were the main control on sediment accumulation at the core site. To make a convincing case would require multiple cores to constrain sediment discharge from the Plata River basin. Any individual core could represent accumulation related to bottom currents or dynamic sediment accumulation along the margin.

Comment #13 - We agree that the second HS2 peak in sedimentation rate and the respective planktonic $\delta^{13}$C anomaly are not perfectly aligned in time. However, this apparent offset can be exclusively due to: (i) the occurrence of [14]C plateaus during HS (e.g., Sarnthein et al., 2007; Franke et al., 2008); and/or (ii) the discretized way our age model was produced in relation to the "continuous" $\delta^{13}$C measurements. Thus, despite of the apparent high temporal resolution showed by the sedimentation rate record of Figure 5i, the information used to produce that curve are the discretized radiocarbon ages.

Choosing ages at slightly different depths may have aligned both records for the second HS2 peak.

Regarding HS3, to the lack of a positive anomaly in sedimentation rates at our core site could be related to the relatively high sea–level during HS3 if compared to the prevailing sea level during HS2 and HS1 (Waelbroeck et al., 2002). The relatively high sea level may have dampened a more significant anomaly in sedimentation rates by shifting the main depocenter towards the continental shelf (e.g., Lantzsch et al., 2014).

To prevent misleading the reader, we added the following sentence to the text.

Page 10, lines 29-31: we added "*However, other factors like shifts in bottom currents and sea level could have also produced the observed changes in sedimentation rates. Detailed age models and more cores from the Rio Grande Cone are necessary to elucidate the main factors controlling the sedimentation rates in that region.*".

Additionally, we emphasized that the "w-structure" is only present in $\delta^{13}C$ for HS3.

Page 10, line 28: we added "*(in this case, only for $\delta^{13}C$)*".

While it is helpful that the authors now include d18O data in the supplemental information, there is no mention of the records here in the discussion. Interestingly, it looks like there is a long-term minimum in d18O at the beginning of HS2, which may be consistent with greater fresh water input to the core site.

Comment #14 - We have not deepened the discussion on the $\delta^{18}O$ records because we believe that: (i) these records do not present clear trends across HS3 and HS2; and (ii) high temporal resolution temperature records from specific depths would be necessary to allow a comprehensive interpretation. Additionally, we mentioned in page 10, line 4 that the $\delta^{18}O$ records are available in the supplementary material.

References

[revised manuscript text omitted]